# Identification of tagged glycans with a protein nanopore

Minmin Li [1,2], Yuting Xiong [1,2], Yuchen Cao[1], Chen Zhang[3], Yuting Li[3], Hanwen Ning[4], Fan Liu[1], Han Zhou[1,3], Xiaonong Li[3], Xianlong Ye[3], Yue Pang [5], Jiaming Zhang[4], Xinmiao Liang[1,3] ✉ & Guangyan Qing [1] ✉

Structural complexity of glycans derived from the diversities in composition, linage, configuration, and branching considerably complicates structural analysis. Nanopore-based single-molecule sensing offers the potential to elucidate glycan structure and even sequence glycan. However, the small molecular size and low charge density of glycans have restricted direct nanopore detection of glycan. Here we show that glycan sensing can be achieved using a wild-type aerolysin nanopore by introducing a facile glycan derivatization strategy. The glycan molecule can induce impressive current blockages when moving through the nanopore after being connected with an aromatic group-containing tag (plus a carrier group for the neutral glycan). The obtained nanopore data permit the identification of glycan regio- and stereoisomers, glycans with variable monosaccharide numbers, and distinct branched glycans, either independently or with the use of machine learning methods. The presented nanopore sensing strategy for glycans paves the way towards nanopore glycan profiling and potentially sequencing.

Glycans, chain-like assemblies of carbohydrates, perform varied and crucial functions in numerous cellular activities[1,2]. Diverse roles of glycans are matched by their highly complex structures, which derive from differences in composition, branching, regio- and stereo-chemistry, and modification[3]. This incomparable structural diversity results in the fact that glycans contain rich potential information that vastly exceeds that of nucleic acids and proteins, which, however, presents a huge challenge to the structural analysis of glycans. Current approaches mainly rely on combined techniques transplanted from genomics and proteomics, including mass spectrometry (MS)[3,4], chromatography[5], and nuclear magnetic resonance (NMR) spectroscopy[6]. However, the progress has lagged far behind, principally because glycan, which undergoes non-template driven biosynthesis, cannot be amplifiable, besides being structurally heterogeneous. And the limitations of the employed techniques are also apparent[7]. For example, MS itself failed to resolve isomers. NMR generally requires high sample consumption (-mg scale) and a considerable amount of analysis time. Recently, several hyphenated tools, for example, ion mobility-MS (IM-MS)[8,9] and cold infrared spectroscopy-MS (IR-MS)[10], have emerged as powerful alternates to advance glycan analysis. However, the resolving power of IM-MS remains to be improved, especially for glycans with large sizes or minor difference[11]. IR-MS depends on custom-built and less accessible instrumentations[12].

Inspired by the nanopore-based single molecule sensing technique that has achieved great success in DNA sequencing[13,14] and been advancing towards protein sequencing[15–17], identifying and sequencing glycans using nanopore has sparked interests[18]. In theory, a glycan

[1]CAS Key Laboratory of Separation Science for Analytical Chemistry, Dalian Institute of Chemical Physics, Chinese Academy of Sciences, Dalian 116023, China. [2]Jiangxi Province Key Laboratory of Polymer Micro/Nano Manufacturing and Devices, School of Chemistry, Biology and Materials Science, East China University of Technology, Nanchang 330013, China. [3]Jiangxi Provincial Key Laboratory for Pharmacodynamic Material Basis of Traditional Chinese Medicine, Ganjiang Chinese Medicine Innovation Center, Nanchang 330000, China. [4]Department of Statistics, Zhongnan University of Economics and Law, Wuhan 430073, China. [5]College of Life Science, Liaoning Normal University, Dalian 116081, China. ✉e-mail: liangxm@dicp.ac.cn; qinggy@dicp.ac.cn

molecule can be sensed as it disrupts the constant ionic current when passing through a nano-sized pore under an applied voltage and thus induces a molecule-specific current blockage signal[19]. Despite an alluring prospect, nanopore sensing of glycans has achieved little progress over the past dozen years. Only a handful of cases that focused on either high molecular weight polysaccharides (e.g., hyaluronic acid[20–22], heparin[23–25], and their fragments[26] with repeating disaccharide units, and xylan[27]) with solid-state and protein nanopores or a few monosaccharides with engineered protein nanopores via a phenylboronic acid-binding strategy[28,29] were reported. However, for smaller but structurally more diverse glycans (*N*- and *O*-linked glycans on glycoproteins, human milk oligosaccharides, etc.) with greater biological significance, single molecule detection with nanopore has not yet been achieved. The key technical hurdles that we face are that the fast passage of glycan through nanopore cannot be sensed because of the small size and weak affinity of glycan with nanopore[30], and that glycan molecules cannot access the nanopore due to the intrinsic electroneutrality of most glycans[31], which seriously challenges the current nanopore sensing strategies.

In this work, we present a solution to this challenge by introducing an aromatic-type tag to glycans via a high-efficiency and facile chemical derivatization to achieve glycan sensing with aerolysin (AeL) nanopore (Fig. 1, a, b), which allows for the unambiguous identification of different glycan isomers, glycans with varying lengths, and branched glycans.

## Results

### Glycan derivatization for nanopore detection

Analytes available for nanopore detection are typically electrically charged. Thus, to probe glycan detection using nanopore, we first chose the negatively-charged sialylglycans to perform the nanopore test. A recombinant wild-type (WT) AeL protein nanopore with the narrowest pore size of ~1.0 nm is employed here (Supplementary Fig. 1)[32], since the small pore can better match small glycan molecules in size. We find that a trisaccharide, 6′-sialyllactose (6SL), that was added into the *cis* solution of AeL nanopore failed to cause the identifiable current blockage signal under +100 mV (Fig. 1c). A similar phenomenon also happens to 3′-sialylactose (3SL). And the situation was not changed by increasing the applied voltage or salt concentration of the electrolyte (Supplementary Fig. 2, Supplementary Note 1). When pentasaccharide sialyllacto-*N*-tetraose a (LSTa) was tested, the expectant signal has not yet emerged (Supplementary Fig. 3). These observations suggest that AeL nanopore cannot sense these small sialylated oligosaccharides, probably due to the small size of these glycans and/or the weak interactions with the nanopore interface[33].

Thus, how to enhance the interaction of glycan with nanopore has become the key to nanopore sensing. Given the highly positively charged characteristic of key sensing regions of AeL pore lumen[34] and the widely used derivatization strategy of the glycan analysis field[35], we next try to chemically link an aromatic (Ar) molecule to glycan and employ the potential cation–π interaction to increase the interaction

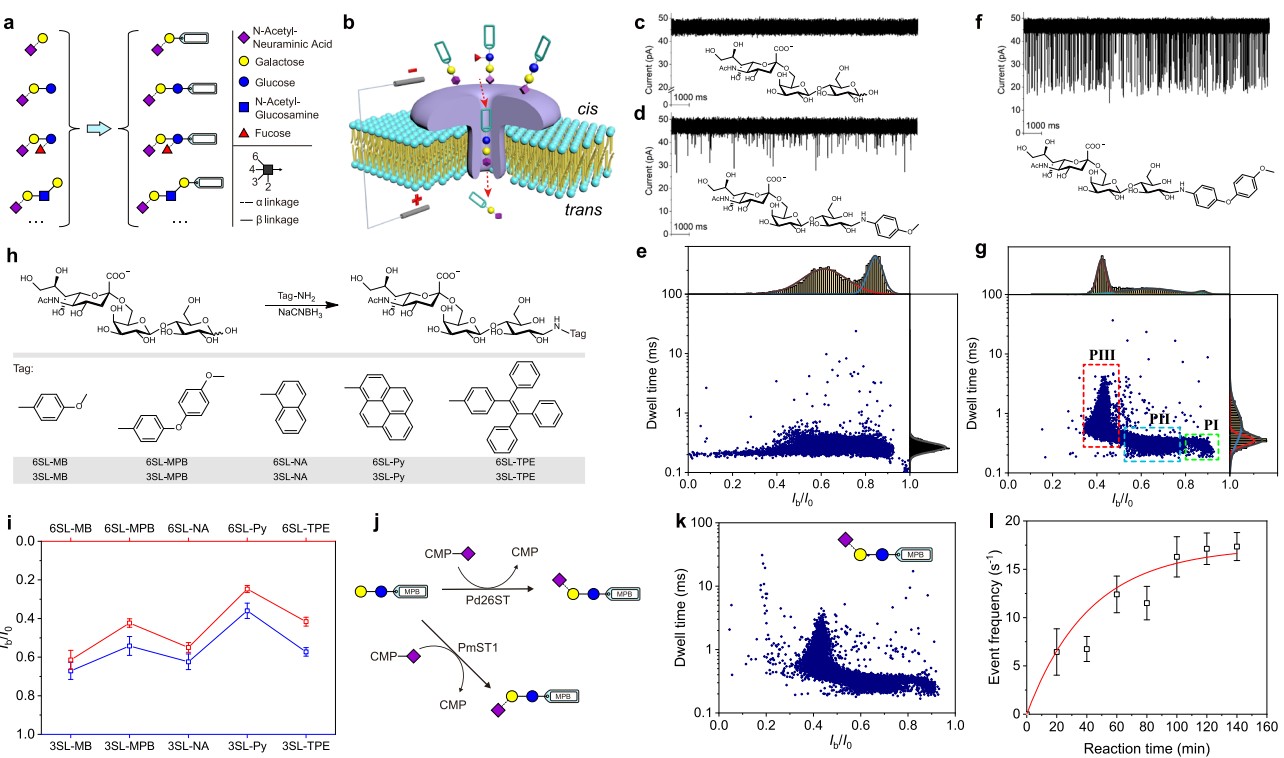

**Fig. 1 | Nanopore-based glycan detection. a** Schematic showing derivatization of glycans depicted with Symbol Nomenclature for Glycans (SNFG)[51]. **b** Schematic of translocation of tagged glycans through AeL nanopore. Representative ionic current traces from 6′-sialyllactose (6SL) (**c**), 1-methoxybenzene (MB)-tagged 6SL (6SL-MB) (**d**), 1-methoxy-4-phenoxybenzene (MPB)-tagged 6SL (6SL-MPB) (**f**), and event scatter plots of 6SL-MB (**e**) and 6SL-MPB (**g**). The populations of PI, PII and PIII in the scatter plot of **g** are shown by the colored boxes. The $I_b/I_0$ and logarithmic dwell time were fitted to multiple or single peak Gaussian distributions. **h** Schematic of glycan derivatives with different tags and the corresponding short names. **i**, $I_b/I_0$ values of the characteristic populations of glycan (6SL and 3SL) derivatives with different tags, including MB, MPB, naphthyl (NA), pyrene (Py), and tetraphenylethylene (TPE). Each data represents the mean value of $I_b/I_0$, and the

corresponding error bar represents the standard deviation (SD) of three independent experiments. The sampling rate of nanopore test of 6SL-Py is 100 kHz. **j** Schematic of the enzymatic synthesis of sialylglycans from MPB tagged lactose (Lac) by sialyltransferases Pd26ST and Pm23ST in the presence of coenzyme cytidine 5′-monophosphate (CMP)-sialic acid. **k** Scatter plot from sialylation reaction product of Pd26ST. **l** Event frequency from the nanopore test of sialylation product *vs.* enzymatic reaction time. Data points are presented as mean ± SD of event frequencies from three independent experiments. All measurements were done in a 10 mM Tris-HCl buffer containing 1 mM EDTA and 1 M KCl with pH 8.0 at +100 mV voltage. Unless otherwise stated, all nanopore data were recorded using a 250 kHz sampling rate with a 5 kHz low-pass filtering. Each scatter plot contains at least 9000 events. Source data are provided as a Source Data file.

of glycan with the nanopore interface. We found that, when 6SL was tagged with 1-methoxybenzene (MB) by a high-efficiency amination reaction with *p*-anisidine, the obtained derivative 6SL-MB can produce obvious blockage signals (Fig. 1d) when added into the *cis* side of AeL with a final concentration of ~ 2 μM. Similarly, 3SL derivative (3SL-MB) also elicited the identifiable blockages (Supplementary Fig. 4), but the amplitudes are relatively smaller than those of 6SL-MB. This indicates that our derivatization strategy makes sialylglycans available for AeL detection. Then the blockage events were processed to extract the current blockage ratio $I_b/I_0$ ($I_b$ and $I_0$ denote the blockage current and open current, respectively) and the dwell time[36], allowing us to characterize the blockage events statistically. The scatter plots of $I_b/I_0$ *vs.* dwell time reveal the similar dwell time and the slight difference in $I_b/I_0$ distributions between 6SL-MB and 3SL-MB (Fig. 1e, Supplementary Fig. 4).

Next, we attempt to optimize the tag molecule to further strengthen the nanopore blockage signals. We found that, when the tag was changed to 1-methoxy-4-phenoxybenzene (MPB), 6SL derivative−6SL-MPB can produce much more obvious current blockages in AeL (Fig. 1f), implying the large-sized tag molecule can cause stronger blockages. And the scatter plot of 6SL-MPB shows a characteristic population (PIII) significantly differing from that of 6SL-MB (Fig. 1, e, g, Supplementary Note 2). Similarly, 3SL derivative−3SL-MPB also displays strong blockages relative to those from 3SL derivative with a MB tag (Supplementary Fig. 5). The characteristic population in scatter plot can be used to identify 6SL and 3SL derivatives. Accordingly, 6SL-MPB is characterized by a mean $I_b/I_0$ value of 0.42 and a mean dwell time of 0.93 ms, while 3SL-MPB can be marked with a mean $I_b/I_0$ value of 0.54 and a mean dwell time of 0.88 ms. These results suggest that the large-sized tag can substantially magnify the difference among glycan isomers, thus helping to achieve the clear identification of glycan linkage isomers. Then, a control experiment towards glycans with a straight-chain alkane as the tag revealed that no identifiable blockage was observed (Supplementary Fig. 6). This further confirmed the critical role of Ar tag, which enables AeL nanopore to sense glycan derivatives, probably relying on the enhanced affinity.

Further, we tested glycans with different Ar tags, for instance, naphthyl (NA), pyrene (Py), and tetraphenylethylene (TPE) (Fig. 1h). Combined, we found that the blockage amplitude varies by tag (Fig. 1i, Supplementary Figs. 4, 5, 7–9), and the dwell time shows a tag size-dependent increase (Supplementary Fig. 10). For example, TPE tagged 6SL (6SL-TPE) produces a mean dwell time of 2.22 ms, which is much longer than those with other tags. Interestingly, α2–6 linked sialylglycan always produced the stronger blockage and longer dwell time than the corresponding α2–3 linked isomer, probably because the sialylglycan with bent α2–6 linkage possesses a larger radical size. Of note is that Py-tagged glycans present much more complex event populations (Supplementary Fig. 8), owing to the easy-to-aggregation feature of planar Py molecular under π−π interaction[37]. TPE-tagged glycans are peculiarly prone to produce blockage with long dwell time and multi-level events (Supplementary Fig. 9), probably because the large size of TPE caused the serious jam of the nanopore. Overall, these results confirm that glycan derivatization with a suitable Ar tag can elicit strong nanopore signals, thus providing a feasible strategy to sense glycans with AeL nanopore.

Given the clear and identifiable blockage signal, the tag MPB is employed to derivatize glycans in the following tests. Prior to that, to probe whether the blockage derived from glycan translocation, an experiment that involves multiple AeL nanopores in a lipid bilayer membrane and the addition of 6SL-MPB of ~100 μM in the *cis* solution was designed (Supplementary Fig. 11, Supplementary Note 3), which lasts for around 8.5 h under +100 mV. The presence of 6SL-MPB was confirmed by an MS analysis of the collected *trans* solution, demonstrating the glycan's translocation from the *cis* to *trans* side through the AeL. Moreover, the translocation was confirmed by the voltage-

dependent variations in event frequency, dwell time, and $I_b/I_0$ (Supplementary Fig. 12, Supplementary Note 4). In addition, our derivatization strategy also demonstrate the viability of sialylated disaccharide detection, where the cognizable and distinguishable nanopore blockages from MPB-tagged 6′-sialylgalactose (6SG-MPB) and 3′-sialylgalactose (3SG-MPB) were observed (Supplementary Fig. 13).

The successful nanopore recording has sparked our interest to monitor the sialylation reaction catalyzed by sialyltransferase. To this end, an MPB-tagged lactose (Lac-MPB) was synthesized as the substrate. Sialylation occurred upon addition of α2–6-sialyltransferase (Pd26ST) into the substrate solution containing coenzyme cytidine-5′-monophospho-*N*-acetylneuraminic acid (CMP-Sia) (Fig. 1j). Since Lac-MPB, CMP-Sia, and the byproduct CMP cannot cause blockage (Supplementary Fig. 14), the obvious blockages observed from the added reaction solution after a period of reaction are attributed to the sialylated product (i.e., 6SL-MPB), and the event distribution matches well with that of 6SL-MPB (Fig. 1k). Besides, the event frequency increased over reaction time (Supplementary Fig. 15). Considering the linear dependence of event frequency on analyte concentration[38,39], the time-dependent event frequency can serve as an indicator of Pd26ST activity assay (Fig. 1l). Similarly, sialylation from α2–3-sialyltransferase (PmST1) can also be detected (Supplementary Fig. 16). Thus, this nanopore sensing offers a highly sensitive method for sialyltransferase activity assay.

## Identification of glycan isomers, glycans with varying lengths and branches

Glycan isomerization that derives from stereochemistry of the monosaccharide building block and regiochemistry of the glycosidic bond has long been a nightmare for MS analysis, thus calling for a novel and readily accessible analytical tool to unambiguously distinguish various isomers[40]. The previous section has indicated that the nanopore data can function as a fingerprint, by which a pair of glycan linage isomers (6SL-MPB and 3SL-MPB) can be clearly distinguished. Thus, we next examine the discerning ability of AeL nanopore towards their mixture. Figure 2a shows the respective event distributions of 6SL-MPB and 3SL-MPB. When their equimolar mixture was tested, the corresponding event distribution displays an obvious stack phenomenon (Fig. 2b). Then we try to isolate the two isomers from the blended nanopore data and determine their respective weights in the mixture by comparing the empirical distribution functions of events and optimizing the Jensen-Shannon divergence (JSD) (Supplementary Fig. 17). For the equimolar mixture, the algorithm procedure reported a weight of 51.55% for 3SL-MPB (Supplementary Fig. 18), showing high accuracy. Then we further measured the mixtures containing 20%, 40%, 60%, and 80% of 3SL-MPB, respectively. According to the nanopore data, the predicted weights are 18.94%, 44.93%, 56.45%, and 76.96%, respectively. And the data points of the predicted weights against the actual molar ratios are closely scattered around the line of $y = x$ (Fig. 2c). This simple binary mixture experiment indicates that nanopore sensing, by combining the applicable algorithm, has the potential to precisely determine glycan relative proportion in a mixture.

Apart from isomerization, resolving the monosaccharide number difference of glycans also is a key precondition to nanopore-based glycan characterization and sequencing. Given this, we turn our attention to a set of linear glycans with varying numbers of monosaccharide building blocks. On the basis of the aforementioned nanopore tests of disaccharide (3SG-MPB) and trisaccharide (3SL-MPB), we further tagged tetrasaccharide (STetra2) and pentasaccharide (LSTa) with MPB and conducted the nanopore tests towards the derivatives STetra2-MPB and LSTa-MPB. Among the four glycans, varying monosaccharide numbers (i.e., chain lengths) are the most striking distinction (inset of Fig. 2d). The data comparison of four glycans reveals that the characteristic blockage enhanced, or in other words, $I_b/I_0$ reduced, with the increase of glycan length (Fig. 2d), while

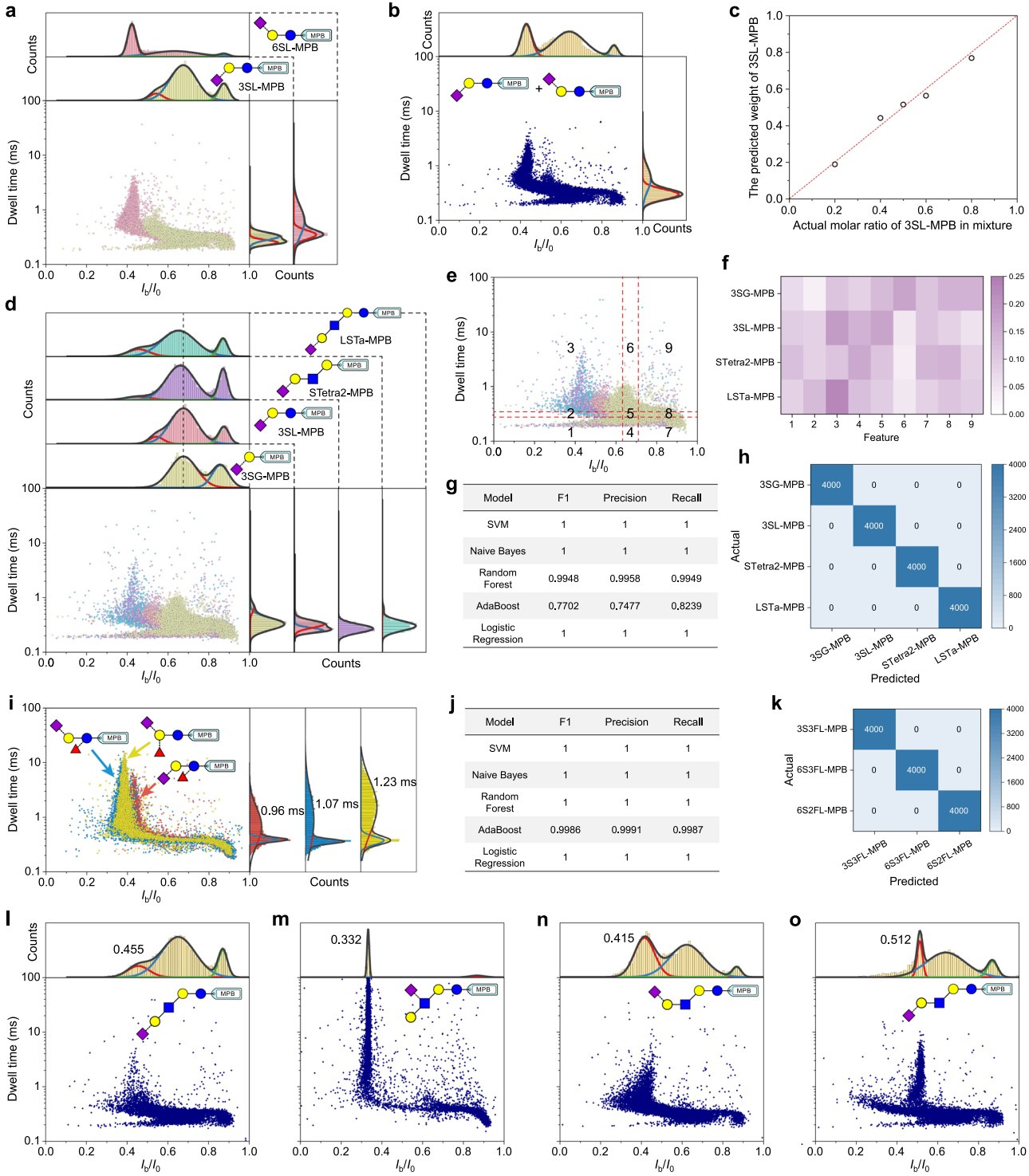

**Fig. 2 | Identification of different glycans. a** Superimposed scatter plots and the corresponding $I_b/I_0$ and dwell time distributions of 3SL-MPB and 6SL-MPB. **b** Scatter plots and the corresponding $I_b/I_0$ and dwell time distributions of the equimolar mixture of 3SL-MPB and 6SL-MPB. **c** The predicted weights of 3SL-MPB against the actual molar ratios in mixtures. **d** Superimposed scatter plots and the corresponding $I_b/I_0$ and dwell time distributions of MPB-tagged disaccharide (3SG-MPB), trisaccharide (3SL-MPB), tetrasaccharide (STetra2-MPB), and pentasaccharide (LSTa-MPB). **e**, Equal frequency binding of the scatter plot from all events of four glycans. **f** The normalized heat map of 9 features of four glycans. **g** Evaluation results of five models in terms of F1, Precision, and Recall scores. These scores were the mean values of 10 replicates, each replicate consists of 100 training and test cycles. **h** Confusion matrix of SVM model from the accumulated prediction result of 4000 test sets after 10 replicates. One training and test cycle includes the

prediction result of 4 test sets of feature data for each glycan. **i** Superimposed scatter plots and the corresponding dwell time distributions of 3S3FL-MPB (red), 6S3FL-MPB (blue), and 6S2FL-MPB (yellow). The characteristic populations were marked with the mean dwell time from Gaussian fit. **j** Evaluation results of all models. **k** Confusion matrix of SVM model from the accumulated prediction result of 4000 test sets after 10 replicates. **l–o** Scatter plots and the corresponding $I_b/I_0$ distributions of MPB-tagged pentasaccharides (LSTa-MPB, LSTb-MPB, LSTc-MPB, and LSTd-MPB). The sampling rate of **m** is 100 kHz. All measurements were done in a 10 mM Tris-HCl buffer containing 1 mM EDTA and 1 M KCl with pH 8.0 at +100 mV voltage. Unless otherwise stated, all nanopore data were recorded using a 250 kHz sampling rate with a 5 kHz low-pass filtering. Each scatter plot contains at least 9,000 events. Source data are provided as a Source Data file.

the dwell time did not show any length-dependent change. Inspection of the scatter plots also displays the difference among the four glycans. However, the large overlap in scatter plots leads to the difficulty of unambiguously identifying glycans using human eye, particularly when it comes to the large number of analyte samples. To achieve the unequivocal identification of analytes according to nanopore data with subtle differences, machine learning-based methods have been increasingly explored as powerful supports[41]. Typical machine learning methods are the employment of various classification algorithms that are used to discriminate and identify different analytes with minor difference in structure[24,42], size[43], or charge[25] depending on the feature data extracted from either the waveforms[24] or the scatter plots[25]. Here, we attempt to exploit the machine learning-based classification approach to identify glycans based on the scatter plots of $I_b/I_0$ $vs.$ dwell time[25]. Firstly, all events of four glycans were plotted together in terms of $I_b/I_0$ $vs.$ dwell time and divided into 9 bins using the equal frequency binning along each axis (Fig. 2e). The obtained boundaries can be used to divide all events of each glycan, thus generating 9 features by normalizing all event numbers in each bin (Supplementary Fig. 19). The heat map of all features indeed displays a stark difference among glycans (Fig. 2f), which suggests that the extracted features can be used to indicate four glycans. Then all events of each glycan were divided into many subsets with equal event number to extract the feature set. The resulting feature matrix was randomly split into two sets: 80% of data as a training set, and the remaining 20% was used as a test set (Supplementary Fig. 20). The accumulated evaluation results for all the test sets show that most models gave superior accuracy scores (Fig. 2g), and most test sets were predicted accurately (Fig. 2h, Supplementary Fig. 21). These results indicate that nanopore sensing with the help of machine learning can unequivocally identify these glycans with single-monosaccharide differences, demonstrating the potential of the combination of nanopore with machine learning to achieve the glycan profiling and even sequencing.

Branching is one of the prevalent structural features of glycans, and thus identifying the branching pattern is also a key part of glycan analysis. Here we prepared three MPB tagged sialylglycans containing branched fucose: 3′-sialyl-3′-fucosyllactose (3S3FL-MPB), 6′-sialyl-3′-fucosyllactose (6S3FL-MPB), and 6′-sialyl-2′-fucosyllactose (6S2FL-MPB), which are three typical regioisomers, and then performed nanopore tests (Supplementary Fig. 22). Figure 2i shows the super-imposed fingerprints from these three glycans. The characteristic populations in the $I_b/I_0$ dimension of 3S3FL-MPB, 6S3FL-MPB, and 6S2FL-MPB are centered at 0.43, 0.37, and 0.38, respectively (Supplementary Fig. 23). The $I_b/I_0$ value of 3S3FL-MPB is larger than that of the other two glycans, which confirmed again that the blockage amplitude from α2−3 linked sialylglycan is smaller than that from α2−6 linked isomers. Besides, they also differ obviously in dwell time, with mean values of 0.96, 1.07, and 1.23 ms for 3S3FL-MPB, 6S3FL-MPB, and 6S2FL-MPB, respectively (Fig. 2i). 6S2FL-MPB shows the longest duration, suggesting the glycan with a α1−2 linked branched fucose is more likely to cause the long residence in AeL nanopore. Then we adopt the previous machine learning method to identify three glycan regioisomers. We find that all five models report the super high accuracy (Fig. 2, j, k, Supplementary Fig. 24). This result shows the potential of nanopore sensing for identifying branched glycans.

To explore the identification ability of AeL nanopore to larger glycans, we further carried out the nanopore tests towards three LSTa isomers, LSTb, LSTc, and LSTd with the same derivatization. The scatter plot of $I_b/I_0$ $vs.$ dwell time of each glycan exhibits the unique pattern (Fig. 2l−o). LSTa and LSTb share the same Galβ1−3GlcNAcβ1−3Galβ1−4Glc skeleton but differ in the linkage and region of sialic acid. In contrast to LSTa (Fig. 2l), LSTb produced a more narrow distribution in $I_b/I_0$ with a much longer duration (Fig. 2m, Supplementary Figs. 25 and 26b), primarily because LSTb adopts a structure with a large size in the axial direction resulting from the sialic

acid that links to the sub-outermost GlcNAc. LSTc and LSTd are typical regioisomers, where the terminal sialic acid is connected to the same Galβ1−4GlcNAcβ1−3Galβ1−4Glc skeleton through either α2−6 or α2−3 linkage. LSTc with an α2−6 linked sialic acid generated much stronger current blockage (Fig. 2n) than LSTd with an α2−3 linked sialic acid (Fig. 2o). Besides, LSTd presents a well distinguished population with a relatively long duration in the scatter plot (Fig. 2o), probably due to the molecular configuration of LSTd that caused a strong interaction with the nanopore interface. Given the distinct patterns in their scatter plots, these pentasaccharide regioisomers can be distinguished. And these distinct differences also render these regioisomers basically distinguishable in the mixture measurements based on the same AeL nanopore (Supplementary Fig. 26, Supplementary Note 5).

## Nanopore detection of neutral glycans

Except for sialylglycans, the presence of abundant neutral glycans further challenges the nanopore sensing since they are not amenable to electrophoretic modulation[31]. Given the success in sensing sialylglycans, we attempt to apply the derivatization strategy to neutral glycans. The first solution that came to mind was the introduction of an anionic substituent-containing an Ar tag. Accordingly, 4-biphenylcarboxylic acid, 1-naphthalenesulfonic acid, 4-azobenzenesulfonic acid, 3-(4-aminophenyl)propionic acid, and 4-(4-aminophenyl)butyric acid were chosen to serve as the tags of the neutral disaccharide lactose. However, all lactose derivatives with these tags failed to induce any evidential nanopore current blockages, suggesting that this derivatization strategy based on these types of tags might not be applicable for neutral glycan sensing. Thus, additional approaches were needed. Since the sialylglycan derivative can cause the current blockages, it is conceivable that a neutral glycan will generate blockages if it is linked to the tag end of a sialylglycan derivative. Figure 3a illustrates the strategy, where 6SL will serve as a carrier to enable capture by AeL and the central Ar unit is responsible for increasing interaction with the nanopore. Based on this, 6SL was first tagged with a 4,4′-diaminodiphenyl ether. Then the resulting 6SL derivative (6SL-DPE-NH$_2$), which proves to be capable of inducing nanopore blockages (Supplementary Fig. 27), as a composite tag, was further linked to the reducing end of Lactose. The obtained Lactose derivative (Lac-DPE-6SL) indeed triggered obvious current blockages in AeL nanopore (Supplementary Fig. 28). And the dwell time of the characteristic distribution differs from that of the composite tag itself. Similarly, we also labeled cellobiose (Cel-DPE-6SL) and maltose (Mal-DPE-6SL) and carried out the nanopore tests. Due to the minute structural difference in the C4 or C1 stereochemistry among the three disaccharides (Fig. 3b), their scatter plots exhibit similar distribution (Supplementary Fig. 28). Nevertheless, we can still observe distinct distributions in their dot density maps (Fig. 3c). Based on this, an improved machine learning-based classification method for discriminating among three isomers still reported an accuracy of up to 96.34% (F1 score) (Supplementary Fig. 29), suggesting the successful discrimination of these neutral epimeric disaccharides.

In addition, we further perform nanopore tests towards two branched neutral glycans, Lewis A (LeA) and Lewis X (LeX) trisaccharide with the same tagging steps. Both two trisaccharide derivatives (i.e., LeA-DPE-6SL and LeX-DPE-6SL) produced quite obvious blockage signals (Supplementary Fig. 30). By comparing the scatter plots and histograms of $I_b/I_0$ or dwell time, we can observe that LeA-DPE-6SL produced a longer blockade duration than LeX-DPE-6SL, giving a dwell time of approximately 1.01 ms from the characteristic population (Fig. 3d). However, in contrast, LeX-DPE-6SL shows a small $I_b/I_0$ value, suggesting that LeX-DPE-6SL caused the stronger blockage in AeL nanopore (Fig. 3e). Moreover, a pair of neutral tetrasaccharides lacto-N-neotetraose (LNnT) and lacto-N-tetraose (LNT) were also tested with AeL nanopore after derivatization. Both glycans induced the prominent blockage events (Supplementary Fig. 31). Blockage event

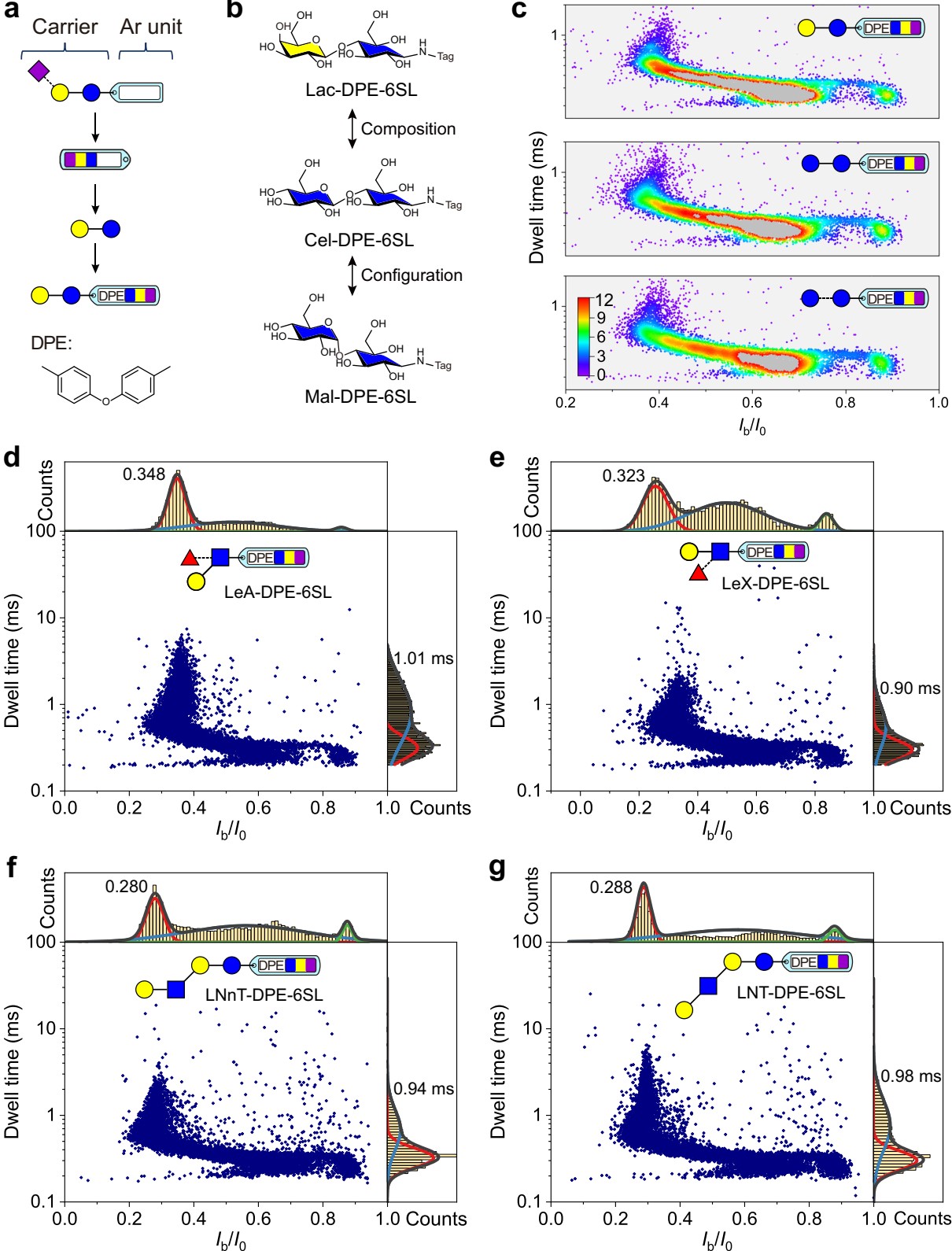

**Fig. 3 | Nanopore detection of neutral glycans. a** Schematic of the derivatization strategy of neutral glycan (Lac, for example). The complex label contains 6SL as a carrier and diphenyl ether (DPE) as the aromatic (Ar) unit. **b** Three complex label tagged disaccharide epimers (Lac-DPE-6SL, Cel-DPE-6SL, and Mal-DPE-6SL) that differ only in the C4 or C1 stereochemistry of the terminal monosaccharide. **c** Dot density maps of nanopore data of three disaccharide derivatives. Scatter plots and the corresponding $I_b/I_0$ and dwell time distributions of two complex label tagged branched trisaccharide isomers (LeA-DPE-6SL and LeX-DPE-6SL) (**d**, **e**) and tetra-saccharide isomers (LNnT-DPE-6SL and LNT-DPE-6SL) (**f**, **g**). All measurements were done in a 10 mM Tris-HCl buffer containing 1 mM EDTA and 1 M KCl with pH 8.0 at +100 mV voltage. All nanopore data were recorded using a 250 kHz sampling rate with a 5 kHz low-pass filtering. Each scatter plot contains at least 9000 events. Source data are provided as a Source Data file.

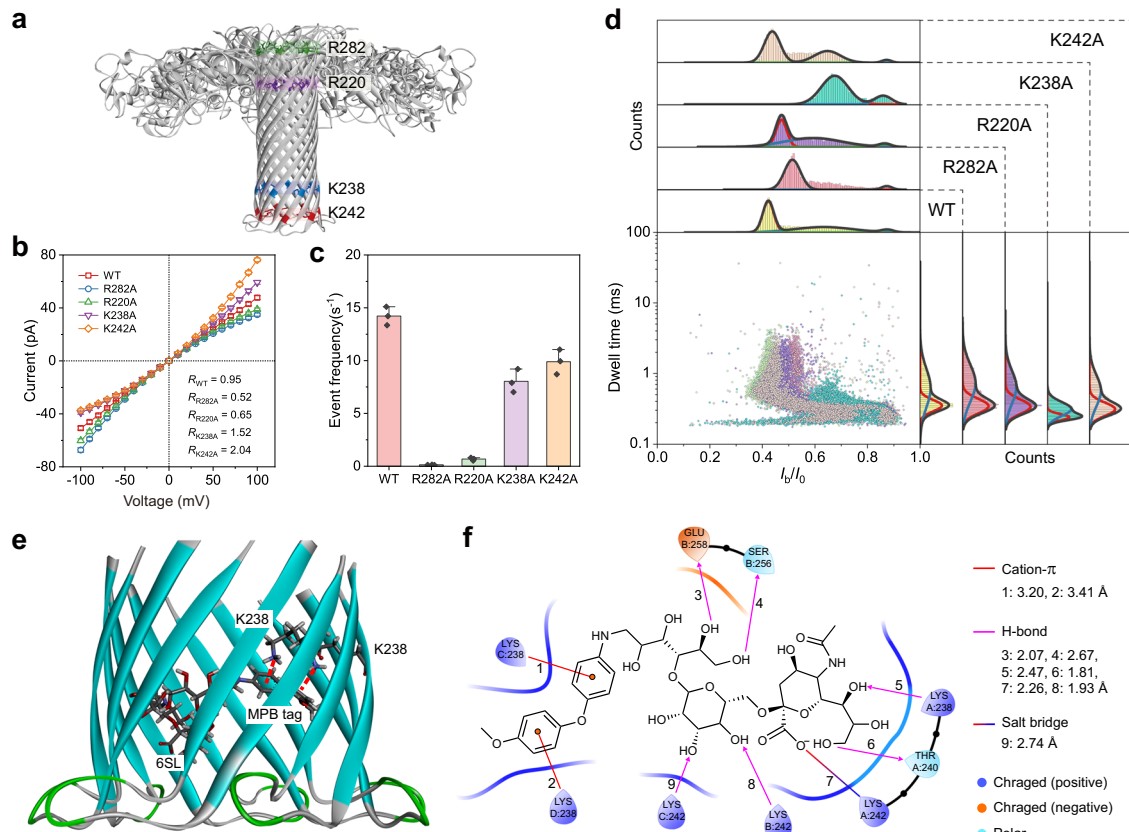

**Fig. 4 | Molecular mechanism of nanopore blockage. a** WT AeL model (PDB: "5JZT") and four residue rings marked with stick models and different colors. **b**, $I-V$ curves of WT AeL and mutants (R282A, R220A, K238A, and K242A), where each data point represents the mean ± SD of three independent experiments. Insets are rectification ratios ($R$) ($|I_{+100\ mV}/I_{-100\ mV}|$). **c** Event frequency of WT AeL and mutants from 6SL-MPB with the same final concentration (2 μM). Data are presented as bar charts with mean values of frequency ± SD of three independent experiments. Individual data points are shown as black dots. **d** Scatter plots and the corresponding $I_b/I_0$ and dwell time distributions of 6SL-MPB measured with WT AeL and four mutants. All measurements were done in a 10 mM Tris-HCl buffer containing 1 mM EDTA and 1 M KCl with pH 8.0 at +100 mV voltage. All nanopore data were recorded using a 250 kHz sampling rate with a 5 kHz low-pass filtering. The scatter plot of R282A contains about 5000 events due to the ultra-low event frequency. Each of the other scatter plots contains at least 9000 events. Partial model (**e**) representing the interaction of 6SL-MPB with AeL in the vicinity of K238 from the docking simulation using Schrödinger software and the detailed interaction pattern (**f**). Source data are provided as a Source Data file.

processing reveals a similar pattern in scatter plots of two glycan derivates (Fig. 3, f, g). However, in contrast with the relatively dispersed blockage events of LNnT derivative (LNnT-DPE-6SL), most blockage events of LNT derivative (LNT-DPE-6SL) centered at a $I_b/I_0$ value of 0.288. And the mean dwell time of the characteristic population of LNnT-DPE-6SL is 0.94 ± 0.17 ms, which is smaller than that of LNT-DPE-6SL (0.98 ± 0.21 ms). Combined, these results suggest that the introduction of a small sialylglycan as the carrier enables the neutral glycan derivative to be captured and sensed by AeL nanopore, achieving nanopore sensing and the differentiation of isomerism at disaccharide, branched trisaccharide, and tetrasaccharide levels. Thus, this strategy provides an inspiration for sensing neutral glycans with nanopore.

## Molecular mechanism of nanopore blockage

It has been established that R282, R220, K238, and K242 residues are key sensing sites of the WT AeL nanopore (Fig. 4a)[34]. To trace the cause of blockage from the glycan derivative, we mutated the four amino acids to alanine through single site-direct mutagenesis respectively, and obtained four mutants: R282A, R220A, K238A, and K242A (Supplementary Fig. 32). As shown in Fig. 4b, compared with WT, K238A, and K242A show more obvious rectifying phenomena, resulting from the symmetry breaking in charge distribution and pore geometry when lysine at the pore exit was replaced with a small neutral alanine. Similarly, mutations at the pore entry (i.e., R282A and R220A)

obviously strengthen the reversed rectifying effect. Then, by using 6SL-MPB to test the mutants, we found that the event frequencies of R282A and R220A dropped sharply to 0.14 and 0.67, respectively, relative to 13.7 from WT (Fig. 4c, Supplementary Fig. 33). Because the positively charged R282 or R220 was removed, the adjacent negatively charged residues (e.g., D216, D222) became prominent and formed an electrostatic barrier at the pore entry, thus preventing the negatively charged glycan from entering. Besides, K238A and K242A also show a slight decrease in event frequency. This phenomenon might be explained by the fact that the mutation at the pore exit indirectly decreases the size of the pore entry and thus increases the difficulty of capturing glycan[44]. Similar phenomena were also observed from the evaluation results of 3SL-MPB and 6SL-NA (Supplementary Figs. 34, 35).

Despite the great difficulty in entering pore, the event characteristics of 6SL-MPB measured by R282A and R220A still resemble those of WT (Fig. 4d). Thus, we reason that R282 and R220 residues function to balance the electrostatic potential of pore entry and allow the access of negatively charged glycans. Further inspection reveals that K238A produced substantially weaker blockages, while K242A caused blockages that are comparable to WT. This suggests the pivotal role of K238 residue in sensing glycan derivative, which was also confirmed by the test results from 6SL-NA (Supplementary Fig. 35). Besides, the mutation of K238 to alanine also significantly impaired the resolving power of AeL towards glycan isomers (Supplementary Fig. 36).

Therefore, these results indicate that the K238 residue ring dominated the sensing ability of AeL towards glycan derivatives.

To further reveal the role of K238 residues, we performed a simulation on the interaction of the K238 zone with 6SL-MPB via an automated molecular docking method. The docking model with the highest score clearly shows two cation–π pairs between K238 residues (from Chains C and D) and Ar groups of the MPB tag (Fig. 4e)[45]. Meanwhile, multiple hydrogen (H)-bonds between saccharide hydroxy groups and several amino acid residues, as well as a salt-bridge between the carboxyl group of sialic acid and a K242 residue (Chain A), are also observed (Fig. 4f), suggesting the possibility of multiple interactions. Thus, we can infer that, during the translocation of glycan derivative with an Ar tag, strong cation–π interactions acting on the Ar tag from K238 residues in the narrowest constriction of AeL pore retard the translocation of glycan derivative. And the retardation might be strengthened by H-bonding and/or salt-bridge interactions, finally resulting in a blockage that can be sensed by AeL nanopore. In addition, a docking pose of 6SL-MPB with the pore entry, where the glycan end has inserted into the pore, predicted the salt-bridge interaction between R282 residue and sialic acid (Supplementary Fig. 37). This pose might explain the translocation direction of the glycan derivative from the sialic acid end.

## Discussion

With the gradual maturity of nanopore DNA sequencing and the continuous advance of nanopore-based protein or peptide sequencing, glycans have been emerging as the latest high-profile analyte of nanopore sensing[18]. In the context of the continuous call for a new widely applicable tool for glycan analysis, nanopore sensing with single- molecule sensitivity provides a highly competitive method capable of dealing with glycan diversity and low abundance. Towards the small oligosaccharides that are at the very core of glycomics analysis, we presented here a derivatization strategy for glycan detection with AeL nanopore. Our strategy addressed the issue that the wild-type AeL nanopore is incapable of sensing the translocated glycan molecules due to the small size of oligosaccharide glycans and the low interaction affinity with the nanopore interface. This strategy demonstrated its potential to identify diverse glycan isomers, glycans with different monosaccharide numbers, and branched glycans. To the best of our knowledge, this is the first successful exploration of nanopore sensing towards structurally complex oligosaccharide glycans, which pushes the boundary of nanopore sensing beyond its traditional focus on nucleic acid and protein and activates its power in the glycomics and glycoscience fields. Thus, this nanopore sensing method provides a highly sensitive and easily accessible analytical tool for glycan profiling. More importantly, single-molecule nanopore recording towards glycans with varying structures represents an important step for nanopore glycan sequencing. A strategy that is available now can rely on the sequential cleavage of a glycan with the known exoglycosidases from the non-reducing end and the subsequent test towards residue glycan. The cleaved monosaccharide type and linkage correspond to the specific exoglycosidase. To better achieve these goals, the following work will focus on improving the resolving power of nanopore by either nanopore protein engineering or chemical modification to achieve label-free detection of native glycans.

Most notably, the introduction of machine learning method in glycan identification experiments has significantly strengthened the identification ability of nanopore. This proof-of-concept experiment based only on the scatter plot pattern foreshadows the great potential of the integration of nanopore sensing with machine learning method in the future glycan analysis field. On the one hand, as nanopore technique advances, both the number of glycan samples with similar structure, including small individual monosaccharide units, that need to be discerned, and the difficulty of artificial identification by nanopore data will inevitably increase. In this regard, machine learning method can be expected to assist nanopore sensing in achieving the unequivocal and rapid glycan identification from a small amount of nanopore data. On the other hand, our proof-of-concept experiment involves multiple isolated steps, including nanopore recording, signal processing, and machine learning-based classification. Given the excellent data processing ability of machine learning[46], the future integration of nanopore sensing with machine learning should be a fusion on a deeper level. The resultant artificial intelligent nanopore can be expected to directly and accurately recognize glycan analytes by identifying the waveform from the pristine current traces immediately after nanopore recording[47].

## Methods

### Preparation of recombinant wild type proaerolysin

According to the reference sequence (GenBank accession number P09167, https://www.ncbi.nlm.nih.gov/protein/113485/), the gene of proaerolysin was expressed and purified to supply the recombinant proaerolysin. The open reading frame (ORF) of proaerolysin, flanked by EcoRI and HindIII restriction sites, was amplified and subcloned into the pET22b(+) vector with a histidine (His) tag. The recombinant proaerolysin was expressed in Rosetta Blue cells. In brief, recombinant proaerolysin protein expression was induced with 0.5 mM IPTG for 24 h at 16 °C. Subsequently, the cell suspension was sonicated, and the soluble supernatant after protein denaturation was collected and subjected to a Ni-NTA His-Bind resin column. The proaerolysin proteins were then refolded, and the concentration was measured using a Bicinchoninic Acid (BCA) Protein Assay kit (Beyotime, China). The purified protein was then analyzed by 15% SDS-PAGE and stained with Coomassie Brilliant Blue (Sangon Biotech, China). Finally, the purified proteins were stored at −80 °C. Prior to use, trypsin agarose was added to the obtained proaerolysin protein solution to convert the dimer to the monomer by slowly rotating the mixed solution at room temperature for 6 h. The activated proaerolysin was stored at −80 °C for later use.

### Preparation of recombinant proaerolysin mutants

The genes of R220A, K238A, K242A, and R282A proaerolysin were codon optimized and synthesized by GenScript, Inc. using a site-directed mutagenesis kit to introduce point mutations in plasmid DNA templates. Target mutant genes were introduced into the pET28a vector by NdeI and XhoI restriction sites. Then, four constructed plasmids with target genes were transformed into BL21(DE3) E. coli, which allows intracellular expression of the protein with a 6×His tag on the N-terminus. The transformed E. coli strains were cultured in Luria−Bertani medium with 100 μg/mL kanamycin. Target recombinant protein was induced with 0.25 mM IPTG at 18 °C overnight. Four recombinant proteins were expressed as solubles and purified according to the same protocol. In brief, the cell suspension was sonicated, and the soluble supernatant after centrifugation was collected. Subsequently, the target protein in the supernatant was captured by the Ni Sepharose FF resins. The protein was further purified by ion exchange chromatography using a HiTrap Q FF column (GE Healthcare) with linear gradient elution. The purity of protein was confirmed by 12% SDS-PAGE. Finally, the purified proteins were stored at −80 °C. Prior to use, trypsin agarose was added to the obtained proaerolysin mutant solution to convert the dimer to the monomer by slowly rotating the mixed solution at room temperature for 6 h. The activated proaerolysin was stored at −80 °C for later use.

### General procedure of preparation of glycan derivatives

Glycan derivatization was performed with a typical reductive amination reaction[48]. Unless otherwise stated, 1 equiv. glycan, 2 equiv. tag reagent, and 5 equiv. sodium cyanoborohydride (NaBH$_3$CN) were dissolved in 0.2 ~ 1 mL anhydrous methanol. Anhydrous dimethyl sulfoxide (DMSO) was added to help dissolve the insoluble glycan.

Glacial acetic acid was added to the solution with a volume ratio of 2% (v/v). Then the mixture solution was transferred quickly into a 4 mL brown glass vial, fastened with a screw cap. The glass vial was heated in an oil bath of 65 °C overnight. Then the reaction solution was cooled to room temperature. A small amount of water was added to quench the unreacted $NaBH_3CN$. After filtration with 0.22 μm Nylon syringe filter, the target product was purified through pre-parative High Performance Liquid Chromatography (HPLC) with a semi-preparative column (10 mm × 250 mm) packed with 5 μm 100 Å C18-silica gel stationary phase. Separations were performed at room temperature with binary gradient elution (eluent A: ultrapure water ($H_2O$) with 0.1%(v/v) trifluoroacetic acid (TFA), eluent B: acetonitrile ($CH_3CN$) with 0.1 %TFA(v/v)) at a flow rate of 3 mL·min$^{-1}$. Unless otherwise specified, a UV/Vis absorbance detector with the wavelength of 254 nm (Channel 1) was used for detection. After freeze-drying, the collected product was characterized with mass spectrometry (MS). The purity of glycan derivatives was determined by carrying out the peak area analysis of HPLC spectrum, which calculates the percentage of peak area in relation to the total area of peaks. The general chromatographic conditions are: BOSTON Green ODS column (4.6 mm × 250 mm, 10 μm), room temperature, binary gradient elution (eluent A: $H_2O$ with 0.1% TFA(v/v), eluent B: $CH_3CN$ with 0.1 %TFA(v/v)), flow rate of 1 mL·min$^{-1}$. Unless otherwise specified, a UV/Vis absorbance detector with a wavelength of 254 nm was used for detection.

Of particular note is the derivatization of neutral glycans (Fig. 3a). Hence, the sialylglycan, 6SL, was chosen as the carrier molecule, which was modified with 4,4′-diaminodiphenyl ether. The obtained derivative (6SL-DPE-NH$_2$) was used to further react with the reducing end of neutral glycans through the exposed amino group, thus realizing the derivatization of neutral glycan. Two steps of reactions involved the reductive amination reaction.

## General procedure of enzymatic synthesis of sialylglycans

Sialylglycan was synthesized by transferring a sialic acid (Neu5Ac) molecule from the donor CMP-Sia to the terminal galactose unit of the substrate with sialyltransferase[49]. Firstly, a Tris·HCl buffer (50 mM, pH 8.45) solution was prepared. Then a total volume of 100 μL solution containing 10 mM substrate (MPB-tagged oligosaccharide with a galactose on its nonreducing end) and 15 mM CMP-Sia in Tris·HCl buffer was prepared. Then ~0.02 U of sialyltransferase (Pd26ST or PmST1) was added into the substrate solution, which was incubated overnight (unless otherwise noted) in a shaker at 37 °C by agitating at 220 rpm. Then, the reaction was terminated by incubating the reaction mixture in a boiling water bath for 5 min and the mixture was centrifuged to remove any possible particles (10,000 rpm for 5 min). The supernatant was collected for characterization with MS and for the use of the nanopore test by directly injecting the product mixture into the electrolyte solution of the *cis* compartment.

## Assembly of aerolysin nanopore

All nanopore recording experiments were performed as described in the document[50]. A perfusion bilayer chamber and a Delrin perfusion cup with a 150 μm aperture compose the test device. Prior to assembly of the device, both sides of the aperture of the Delrin perfusion cup were painted with the lipid solution (3 mg 1,2-diphytanoyl-*sn*-glycero-3-phosphocholine dissolved in 100 μL n-decane) by using a sable paintbrush (size 00) and allowed to stand for solvent evaporation. Then, both compartments composed of the perfusion chamber and the perfusion cup were injected with 1 mL electrolyte solution respectively. Unless otherwise stated, the electrolyte is 10 mM Tris·HCl containing 1 mM EDTA and 1 M KCl with pH 8.0. Among which, the left compartment (i.e., the perfusion cup) was defined as the *trans* compartment, and the other compartment was defined as the *cis* compartment. A pair of Ag/AgCl electrodes that were connected to the

headstage of a patch clamp instrument and used to apply potential across the lipid bilayer membrane were separately immersed into the electrolyte solution of both the *cis* and *trans* compartments, where the Ag/AgCl electrode in the *cis* compartment was defined as the virtual ground. After adding the lipid solution into the *cis* compartment with a sable paintbrush, a lipid bilayer was formed by lowering the solution level of the *cis* compartment below the aperture and raising the solution level above the aperture using a syringe. Upon the lipid bilayer formation, the ionic current changed to 0 pA, indicating that the aperture was sealed, and the entire circuit was open. Then the aerolysin monomer solution was injected adjacent to the aperture in the *cis* compartment to initiate pore insertion under a positive potential of ~ +200 mV. Upon single pore formation, the potential was dropped to +10 mV, and the solution in the *cis* compartment was stirred gently with a syringe needle to avoid further insertion of more pores.

## Nanopore measurement and data analysis

Assembly of aerolysin nanopore was detailed in Supplementary Information. Once a single aerolysin nanopore formed, the glycan solution was added into 1 mL electrolyte solution (10 mM Tris·HCl containing 1 mM EDTA and 1 M KCl with pH 8.0) of the *cis* compartment with a final concentration of approximately 2 μM (unless otherwise noted). The ionic current traces were recorded using an Axopatch 200B patch clamp amplifier paired with a Digidata 1550B digitizer (Molecular Devices, Sunnyvale, CA, USA). Unless otherwise stated, the applied voltage during all measurements was +100 mV, and all ionic current signals were sampled at 250 kHz and filtered with a low-pass Bessel filter at 5 kHz by running Clampex 11.2 software (Molecular Devices, Sunnyvale, CA, USA) under an experimental environment (22-24 °C of temperature, 40%~60% of humidity). In the test process, the measurement device and the headstage of the patch clamp amplifier were placed in a tailor-made copper box on a vibration isolation table with a Faraday cage. The recorded current traces were processed by using MOSAIC v2.2 software developed by Dr. Balijepalli[36], which is especially good at resolving the short-lived events produced by small oligosaccharide molecules. Then the subsequent data presentation, including scatter plot generation, histogram plotting, and the corresponding fitting were performed on OriginPro 2021 (OriginLab, Northampton, Massachusetts, USA).

On the enzymatically synthesized sialylglycans, the nanopore test was performed by directly injecting the enzymatic reaction solution after a period of reaction into the *cis* solution. To evaluate the enzymatic activity of sialyltransferase, an equal volume of reaction solution was withdrawn and injected into the *cis* solution of the newly assembled AeL nanopore test set-up with the same time interval.

## Glycan identification and weight prediction in mixture

In probability theory and statistics, JSD is a popular method of measuring the similarity between two probability distributions. It is based on the Kullback−Leibler (KL) divergence, with some notable differences. For instance, it is symmetric and it always has a finite value. Thus, JSD can function as one of the distribution comparison techniques. Here, blockage event parameters (i.e., $I_b/I_0$ *vs.* dwell time) from a nanopore test of a glycan can be regarded as a probability distribution. Thus, identification of glycan was achieved by comparing the distribution of measured blockage events to that of the provided ground-truth blockage events in terms of JSD value. A JSD value near zero means that the measured result and ground-truth result are similarly distributed, and thus the glycan measured can be identified.

Supplementary Fig. 18 illustrates the detailed flow of our calculation process. Firstly, suppose there are $N$ types of glycans. Each glycan was measured by AeL nanopore, and the corresponding blockage events were obtained as the ground-truth result. Now let's suppose again that there is a measured mixture sample with an unknow glycan and an unknown proportion that needs to be

determined. After nanopore measurement, the corresponding blockage events were obtained as the measured result.

Given $i \in \{1, 2, \ldots, N\}$, for the $i$-th type glycan, $f_i(x)$ is used to denote the empirical probability density function (EPDF) of all the blockage events (i.e., $I_b/I_0$ vs. dwell time) obtained from the nanopore data of this type of glycan. Then, $f_{mix}(x)$ refers to the EPDF of the measured results from the mixture sample. $w_i$ refers to the objective weight parameter of the $i$-th type molecule in the mixture sample. Then, the projected EPDF $\hat{f}_{mix}(x)$ for the mixture sample is formulated as

$$\hat{f}_{mix}(x) = \sum_{i=1}^{N} f_i w_i(x) \tag{1}$$

Where $\sum_{i=1}^{N} w_i = 1$, and $w_i \geq 0$ for all $i$. The objective is to optimize $w_i$ to make the projected $\hat{f}_{mix}(x)$ be consistent with $f_{mix}(x)$. With the optimal $w_i$, the underlying true weights of each type of glycan in the mixture sample can be inferred resultantly.

To achieve the objective, we adopted the optimization of JSD. JSD associated with $f_{mix}(x)$ and $\hat{f}_{mix}(x)$ is given by

$$\sum f_{mix}(x) \log \frac{f_{mix}(x)}{\hat{f}_{mix}(x)} + \sum \hat{f}_{mix}(x) \log \frac{\hat{f}_{mix}(x)}{f_{mix}(x)} \tag{2}$$

With Formula (1) the optimization problem is constructed as follows

$$\min_{w_1, w_2, \ldots, w_N} \left( \sum f_{mix}(x) \log \frac{f_{mix}(x)}{\hat{f}_{mix}(x)} + \sum \hat{f}_{mix}(x) \log \frac{\hat{f}_{mix}(x)}{f_{mix}(x)} \right) \tag{3}$$

$$s.t. \sum_{i=1}^{N} w_i = 1, w_i \geq 0, \forall i \tag{4}$$

Thus, for the nanopore data the identification of glycans and weight calculation is a conventional convex optimization problem, the optimal solution (i.e., the optimal weights) of which is unique and can be solved trivially by gradient descent method.

## Machine learning-based classification

First, the obtained nanopore signal parameters, $I_b/I_0$ and dwell time, serve as the raw data, which was processed according to requirements by using PyCharm (version 2021.3.3). ML-based classifications were performed using the scikit-learn (version 0.24.2) library. The five ML algorithms used in this work are SVM, Naive Bayes, Random Forest, Adaboost, and Logistic Regression. For the AdaBoost algorithm, we chose Decision Tree Classifier as the base estimator, and the kernel implementation was SAMME.R. The maximum number of weak learners was set to 100. For the Logistic Regression algorithm, we chose the regularization of Ridge(L2) type and size of C = 1.0. Liblinear was chosen for the internal optimization algorithm due to the small size of the current data set. For Naive Bayes, a multinomial Naive Bayes classifier with a Laplace smoothing parameter of 1.0 was selected, because it is suitable for classification with discrete features of this dataset. For the Random forest, the number of trees in the forest was set to 25 when the sample discrimination was large, and it was set to 80 when it met hard-to-classify samples. For SVM, the penalty, kernel, and iteration limits were L2, RBF, and −1 for no limit, respectively.

Before machine learning, features need to be extracted from raw data using feature engineering methods. Inspired by the reported work[25], for four glycans (i.e., 3SG-MPB, 3SL-MPB, STetra2-MPB, and LSTa-MPB with an identical event number of approximately 40,000), all the nanopore events (~160,000 events) were plotted together into a scatter plot ($I_b/I_0$ vs. dwell time). Then the scatter plot was divided into $3 \times 3$ bins by using equal-frequency binning (Supplementary Fig. 20, a, b). Thus, the boundaries were determined. In addition, we attempt to

split all nanopore data of each glycan into many subsets with an equal event number. To ensure the reasonability of dividing, we used the KL divergence to evaluate the similarity between the distribution of each subset events with varied event numbers and overall events. The more similar the two probability distributions are, the smaller the KL divergence is. The mean KL value from the comparison between all subsets with an equal event number and the overall event of a glycan was plotted against the subset size (i.e., the event number of the subset) (Supplementary Fig. 20c). Considering the fluctuation of KL and the number of subsets, all events of each glycan were divided into 20 subsets, each of which contains 2000 events (Supplementary Fig. 20d). Then, the scatter plot of each subset was partitioned into 9 domains with the boundaries above, which therefore provides 9 feature values by counting all the events in each domain and normalizing (Supplementary Fig. 20e). Thus, for each glycan sample, 20 sets of features (each set consists of 9 features) were obtained from 20 event subsets. Accordingly, the feature matrix of four glycans was composed of $20 \times 4$ feature sets.

During the evaluation process, we strictly performed the validation procedure. As shown in Supplementary Fig. 21, the stratified random sampling method was adopted to split the feature matrix. All feature sets from each glycan represent a strata, which was randomly split into an 80% training set and a 20% test set. That is, 16 feature sets were randomly obtained as training sets to the train model based on the selected algorithm. The remaining 4 feature sets, namely 4 test sets, were applied to test the trained classifiers and ensure reliability. Model evaluation metrics contain the scores of F1, Precision, and Recall. After one cycle of training and test process, 4 test sets from each glycan were predicted. This cycle was repeated 100 times. Thus the prediction of 400 test sets from each glycan was completed and the scores of F1, Precision, and Recall were accumulated, which constituted an experiment. Finally, the prediction result of 4000 test sets was obtained after 10 replicates to plot the confusion matrixes of five models (Fig. 2h, Supplementary Fig. 22), and the average scores of F1, Precision, and Recall were also obtained.

Furthermore, for three branched glycans (3S3FL-MPB, 6S3FL-MPB, and 6S2FL-MPB; with an identical event number of approximately 40,000), the ML approach based on the above binning and subset dividing method also reported high accuracy for glycan identification (Fig. 2, j, k, Supplementary Fig. 25). Notably, for three neutral disaccharide derivatives (Lac-DPE-6SL, Cel-DPE-6SL, and Mal-DPE-6SL, with an identical event number of approximately 30,000), we increased the feature number and optimized the subset size to improve the classification accuracy. Based on the feature numbers ranging from $4^2$-$50^2$ and the subset size ranging from 1000 to 3000 events, we performed the classifier performance tuning for each of the selected algorithms to build the best classifier. Supplementary Fig. 30 shows the optimal results of five models according to optimal binning number and subset size.

## Molecular docking simulation

Potential interaction of a glycan derivative with the AeL nanopore interface was investigated using molecular docking based on the Schrödinger software suite (Schrödinger, LLC, New York, NY). The initial structural model of wide type (WT) AeL was retrieved from the Protein Data Bank (PDB ID: "5JZT"). Protein Preparation Wizard tool in Schrödinger suite was used to optimize the protein by removing water molecules and adding the missing hydrogen atoms, side chains, and bond orders. PROPKA program was used to calculate the ionization state of polar amino acids at pH 8.0 and Protassign program was used to optimize the H-bond network. A full-atom energy minimization of the protein structure was carried out using the OPLS4 force field to refine the geometric structure and the previous H-bond network. The glycan derivative (6SL-MPB) was prepared using the LigPrep tool of the Schrödinger suite by converting 2D structure to 3D structure,

followed by the generation of the correct ionizable state at pH 8.0. As for the receptor grids, the grid in the pore entry of AeL was set as 25 Å × 25 Å × 25 Å, the geometric center of the R282 region reserved as the grid centroid. The grid in K238 region was set as 20 Å × 20 Å × 20 Å to ensure the grid can cover the pore exist, the geometric center of the K238 region was reserved as the grid centroid. Docking calculations were completed with Glide docking protocol of SP (standard precision) and XP (extra precision). 20 poses were outputted. Some top-ranking poses were checked manually, the optimal poses were analyzed and outputted with Maestro.

## Statistics and reproducibility

For nanopore-related statistics the number of replicates was provided in the corresponding figure caption. The extracted blockage events with $I_b/I_0$ greater than 0.99 (the nanopore is completely unoccupied) from the MOSAIC software processing were excluded from the statistics for the accurate representation of blockage signal. No further data was excluded from analyses. No statistical method was used to predetermine sample size. No randomization or blinding was used.

## Reporting summary

Further information on research design is available in the Nature Portfolio Reporting Summary linked to this article.

## Data availability

The main data supporting the findings of this study are available within the main text, the Supplementary Information file, and the Source Data files. Additional raw data are available at https://doi.org/10.6084/m9.figshare.22241326.v2. Source data are provided with this paper.

## Code availability

The weight prediction in mixture was done with a custom code "MIX-distribution" that is available at https://doi.org/10.5281/zenodo.7714328. Machine learning was based a custom code "Glycan-Classification" that is available at https://doi.org/10.5281/zenodo.7711079. All data that were used for weight prediction and machine learning classification were accompanied for code validation.

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

## Acknowledgements
This work was financially supported by: National Natural Science Foundation of China (21922411 and 22174138 to G.Q., 21934005 to X.L., 22004120 to M.L., 22104013 to Y.X.), Natural Science Foundation of Jiangxi Province (20212BAB213002) to M.L., DICP Innovation Funding (DICP-RC201801, DICP I202008), Dalian Outstanding Young Scientific Talent (2020RJ01), and National Key R&D Program of China (Grant No. 2020YFC3400800) to G.Q.

## Author contributions
M.L., X.L., and G.Q. conceived and directed the project. M.L. and X.Y. designed the experiments. M.L. and Y.X. performed the glycan derivatization with the help of Y.C., Y.P. prepared the wild type proaerolysin protein. X.Y. prepared the mutant proteins. M.L. and Y.X. completed the nanopore measurements. H.N. and J.Z. performed the glycan weight calculation. M.L., C.Z., F.L., and X.L. carried out the machine learning analyses. Y.L and H.Z. carried out the molecular docking. M.L. wrote the manuscript. X.L., and G.Q. revised the manuscript. All authors participated in the discussion of experiment results and confirmed the final manuscript.

## Competing interests
The authors declare no competing interests.
