## [Peer Review File · Nature Communications]

REVIEWER COMMENTS

Reviewer #1 (Remarks to the Author):

General comments:

Much structural information to elucidate glycan sequences can be accessed already by MS2. The determination of monosaccharide linkage is considerably more difficult by MS2 and the method described by the authors has the potential to become an effective and advantageous advance. However, the only linkage isomers illustrated here as having distinguishable scatter plot patterns are the 3SL vs. 6SL, whereas the differences between lactose, cellobiose and maltobiose are too small to be able to make assignments. These, can only be perceived by dot density maps in Fig 3d, rather than Ib/I0 and dwell time 1d distribution scatter.

A further point is that the method is shown here to be useful only with short glycan isomers, and an application to longer glycans is lacking. The difference in one linkage within a trisaccharide is shown here to generate some visible shifts on scatter plot pattern. Whether this difference can be made to be distinguishable among larger glycans e.g. LSTa, LSTd, LSTc, LSTb or isomers of oligomannose M7N2 from RNaseB remains to be investigated.

Overall, the use of English language needs to be improved.

Some specific comments:

Lines 217-233: the analysis of a branched structure was based on the comparison of the branched sequence Gal4-(Fuc3)-Glc-DPE-6SL with the linear sequence Fuc2-Gal4-Glc-DPE-6SL. These two sequences do not really share structural similarity. A fair comparison would be to compare a pair of similarly branched glycans such as Lea and Lex, or two isomers of a mono-sialylated biantennary N-glycan with the Sia attached to one or other LacNAc branch. Can the authors do this?

Lines 240-242: the labels used for neutral glycans all shared a negatively charged group directly linked to an aromatic ring; these did not work. This can be due to the presence of an additional negative charge in the aromatic system. As the 6SL with the DPE carrier can work a rational design for neutral glycans would be a linker with a carboxylic acid far away from the aromatic ring, for example 3-(4-aminophenyl)propionic acid or 4-(4-aminophenyl)butyric acid. This is really worth attempting

Lines 274-275: LNnT was employed here, it would be good for the authors to show the data for the LNT isomer, in order to observe any differences between the patterns from these two isomers, which are difficult to distinguish using MS.

Reviewer #2 (Remarks to the Author):

The idea of employing pi-electron compounds interacting with nanopore surfaces as tags to recognize glycans at the single molecule level is intriguing. By utilizing tags, methods to identify chain length, branching, and neutrality, which cause difficulties in glycan recognition, have also been proposed. Machine learning of the acquired measurement data enables highly accurate single molecule identification. Furthermore, to comprehend the intermolecular interactions between nanopores and tag molecules, mutated nanopores have been developed and molecularly interpreted. Based on the aforementioned criteria, and after addressing the issues raised throughout the peer review process described below, this manuscript is considered appropriate for this publication.

1. The crucial information, such as an overview of nanopore measurement methods, machine learning approaches, and molecular modeling methods, should be provided in the methods section of the text.
2. Is machine learning required for single molecule detection of glycans? If it is essential, the success of single molecule identification of glycans depends on the integration of hardware (tag introduction)

and software (machine learning). If fusion is necessary, it should be introduced with appropriate papers in the introduction.

3. Lines 136 to 147 To confirm that the substance going through the nanopore is 6SL-MPB, the trans solution should be examined by mass spectrometry or NMR after 5 hours of monitoring. The current writing style leads one to believe that substances other than just 6SL-MPB are also mixed in the solution.

4. Does the numerical representation in the confusion matrix in the text and the Supporting data represent the number of ionic current-time waveforms? You must explain what the numbers in the matrix mean.

5. The numbers in the confusion matrix are different for each of the three sorts of molecules. For instance, in Figure 2h, they are 3,000, 3,000, and 4,000. Usually, in machine learning, the number of data for each glycan is kept constant to prevent bias. Here everyone should converge on 3,000.

General response to Reviewers:

We appreciate all the Reviewers for their valuable comments in improving our manuscript. While all Reviewers provided some positive remarks, each Reviewer also raised a number of concerns, which we have addressed through additional experiments and by revising the description of the text. In the following is our point-by-point response to all specific comments of the Reviewers. The Reviewers' comments are in black, our responses immediately follow in blue color. In the revised manuscript and Supplementary Information, all the revised texts have been highlighted in red color.

Finally, we sincerely thank the three Reviewers for their valuable inputs that helped us make significant improvements in the revised manuscript.

Response to the Reviewer#1-----	Page 2-6
Response to the Reviewer#2-----	Page 7-11
Response to the Reviewer#3-----	Page 12-28

Reviewer #1 (Remarks to the Author):

Much structural information to elucidate glycan sequences can be accessed already by MS². The determination of monosaccharide linkage is considerably more difficult by MS² and the method described by the authors has the potential to become an effective and advantageous advance. However, the only linkage isomers illustrated here as having distinguishable scatter plot patterns are the 3SL vs. 6SL, whereas the differences between lactose, cellobiose and maltobiose are too small to be able to make assignments. These, can only be perceived by dot density maps in Fig 3d, rather than Ib/I0 and dwell time 1d distribution scatter.

Response: We thank the Reviewer for the positive comment to our manuscript, which will urge us to continue move forward in nanopore glycan sensing filed.

We agree with the Reviewer that the minute differences among small glycan molecules are difficult to discern using the scatter plot pattern only with our eyes. And the dot density map in some cases can indeed be used to indicate the minute difference. However, we think the dot density map is the presentation of the scatter plot in the eye of computer. Thus the scatter plots, despite the similar patterns that cannot be discerned with our eyes, could be identified with the computer. This is also why we employed the machine learning approach to identify the glycans. And we firmly believe in the future a computer procedure or software could be developed to identify the glycan with the scatter plot, even directly with the waveforms from the pristine current traces, which is also one of our following works.

A further point is that the method is shown here to be useful only with short glycan isomers, and an application to longer glycans is lacking. The difference in one linkage within a trisaccharide is shown here to generate some visible shifts on scatter plot pattern. Whether this difference can be made to be distinguishable among larger glycans e.g. LSTa, LSTd, LSTc, LSTb or isomers of oligomannose M7N2 from RNaseB remains to be investigated.

Response: We thank the Reviewer for the suggestion. We have added the nanopore experiments for LSTa, LSTb, LSTc, and LSTd derivatives. Results show that all four pentasaccharide isomers can be distinguished with scatter plots (Fig. R1). This further suggests that our nanopore sensing approach can identify those much larger

glycans. We have added the graphics (in Fig.2 l-o) and the related description in the revised manuscript as follows.

“To explore the identification ability of AeL nanopore to larger glycans, we further carried out the nanopore tests towards three LSTa isomers, LSTb, LSTc, and LSTd with the same derivatization. The scatter plot of I_b/I_0 vs. dwell time of each glycan exhibits the unique pattern (Fig. 2l-o). LSTa and LSTb share the same Gal β 1–3GlcNAc β 1–3Gal β 1–4Glc skeleton but differ in the linkage and region of sialic acid. In contrast to LSTa (Fig. 2l), LSTb produced more narrow distribution in I_b/I_0 with much longer duration (Fig. 2m, Supplementary Fig. 26 and 27b), primarily because LSTb adopts a structure with large size in axial direction resulting from the sialic acid that links to the sub-outermost GlcNAc. LSTc and LSTd are typical regioisomers, where the terminal sialic acid is connected to the same Gal β 1–4GlcNAc β 1–3Gal β 1–4Glc skeleton through either α 2–6 or α 2–3 linkage. LSTc with an α 2–6 linked sialic acid generated much stronger current blockage (Fig. 2n) than LSTd with an α 2–3 linked sialic acid (Fig. 2o). Besides, LSTd presents a well discernible population with relatively long duration in scatter plot (Fig. 2o), probably due to the molecular configuration of LSTd that caused a strong interaction with nanopore interface. Given the distinct patterns in their scatter plots, these pentasaccharide regioisomers can be distinguished. And these distinct differences also render these regioisomers basically distinguishable in the mixture measurements based on a same AeL nanopore (Supplementary Fig. 27).”

Fig. R1. Fig. 2. l-o, Scatter plots and the corresponding I_b/I_0 distributions of four pentasaccharide (LSTa, LSTb, LSTc, LSTd) derivatives.

Overall, the use of English language needs to be improved.

Response: We have carefully checked the writing of the manuscript and have revised the wrong and less rigorous presentations. Besides, we have also invited a colleague to read the text and comment on readability. Then, we have made revision

accordingly to improve the readability.

Lines 217-233: the analysis of a branched structure was based on the comparison of the branched sequence Gal4-(Fuc3)-Glc-DPE-6SL with the linear sequence Fuc2-Gal4-Glc-DPE-6SL. These two sequences do not really share structural similarity. A fair comparison would be to compare a pair of similarly branched glycans such as Lea and Lex, or two isomers of a mono-sialylated biantennary N-glycan with the Sia attached to one or other LacNAc branch. Can the authors do this?

Response: We thank the Reviewer#1 for the suggestion. As the Reviewer#1 said, the comparison between a branched 3FL and a linear 2FL seems a little unfair. According to the suggestion, we synthesized Lewis A (LeA) and Lewis X (LeX) triose derivatives by tagging the composite tag and performed the nanopore tests. The results show that this two branched glycan can be distinguished depending on the difference in dwell time and current blockage distribution, as shown in Fig.R2. Accordingly, we have replaced the test results of 3FL and 2F with those of LeA and LeX (in Fig. 3. d and e) and also added the related description in the revised manuscript as follows.

“..., we further perform nanopore tests towards two branched neutral glycans, Lewis A (LeA) and Lewis X (LeX) trisaccharide with the same tagging steps. Both two trisaccharide derivatives (*i.e.*, LeA-DPE-6SL and LeX-DPE-6SL) produced quite obvious blockage signals (Supplementary Fig. 31). By comparing the scatter plots and histograms of I_b/I_0 or dwell time, we can observe that LeA-DPE-6SL produced a longer blockade duration than LeX-DPE-6SL, giving a dwell time of approximately 1.01 ms from the characteristic population (Fig. 3d). However, in contrast, LeX-DPE-6SL shows a small I_b/I_0 value, suggesting that LeX-DPE-6SL caused the stronger blockage in AeL nanopore (Fig. 3e). ...”

Fig. R2. Fig.3. d and e, Scatter plots and the corresponding I_b/I_0 and dwell time distributions of LeA and LeX derivatives from AeL nanopore tests.

Lines 240-242: the labels used for neutral glycans all shared a negatively charged group directly linked to an aromatic ring; these did not work. This can be due to the presence of an additional negative charge in the aromatic system. As the 6SL with the DPE carrier can work a rational design for neutral glycans would be a linker with a carboxylic acid far away from the aromatic ring, for example 3-(4-aminophenyl)propionic acid or 4-(4-aminophenyl)butyric acid. This is really worth attempting.

Response: We thank the Reviewer for the suggestion. By using 3-(4-aminophenyl)propionic acid (APP) and 4-(4-aminophenyl)butyric acid (APB) as tag molecules, we have synthesized lactose derivatives: Lac-APP and Lac-APB. However, AeL nanopore measurements for these two lactose derivatives still did not show evidential current blockage signal. We guess that these derivatives, despite a carboxylic acid that is far away from the benzene ring, are relatively too small to induce the blockage of nanopore.

Lines 274-275: LNnT was employed here, it would be good for the authors to show the data for the LNT isomer, in order to observe any differences between the patterns from these two isomers, which are difficult to distinguish using MS.

Response: We thank the Reviewer for the suggestion. We have carried out the nanopore test for the LNT isomer after a derivation step. Indeed, this pair of isomers show different patterns in scatter plots (Fig. R3). Accordingly, we have included the graphics in Fig.3 (Fig. 3. f and g) and also added the related description in the revised manuscript as follows.

“... Moreover, a pair of neutral tetrasaccharides lacto-N-neotetraose (LNnT) and lacto-N-tetraose (LNT) were also tested with AeL nanopore after derivatization. Both glycans induced the prominent blockage events (Supplementary Fig. 32). Blockage event processing reveals the similar pattern in scatter plots of two glycan derivatives (Fig. 3, f and g). However, in contrast with the relatively dispersed blockage events of LNnT derivative (LNnT-DPE-6SL), most of blockage events of LNT derivative (LNT-DPE-6SL) centered at a I_b/I_0 value of 0.288. And the mean dwell time of the characteristic population of LNnT-DPE-6SL is 0.94 ± 0.17 ms, which is smaller than that of LNT-DPE-6SL ($0.98 \pm$

0.21 ms). ...”

Fig. R3. Fig.3. f and g, Scatter plots and the corresponding I_b/I_0 and dwell time distributions of LNT and LNT derivatives from AeL nanopore tests.

Reviewer #2 (Remarks to the Author):

The idea of employing pi-electron compounds interacting with nanopore surfaces as tags to recognize glycans at the single molecule level is intriguing. By utilizing tags, methods to identify chain length, branching, and neutrality, which cause difficulties in glycan recognition, have also been proposed. Machine learning of the acquired measurement data enables highly accurate single molecule identification. Furthermore, to comprehend the intermolecular interactions between nanopores and tag molecules, mutated nanopores have been developed and molecularly interpreted. Based on the aforementioned criteria, and after addressing the issues raised throughout the peer review process described below, this manuscript is considered appropriate for this publication.

Response: We thank the Reviewer for the positive comments to our manuscript. We have added some additional experiments and made some modifications to the manuscript to address the reviewer's concerns.

1. The crucial information, such as an overview of nanopore measurement methods, machine learning approaches, and molecular modeling methods, should be provided in the methods section of the text.

Response: We thank the Reviewer for the suggestion. We have provided essential experimental details, including the preparation of glycan derivatives, the enzymatic synthesis of sialylated glycans, nanopore measurement and data analysis, glycan identification and weight prediction in mixture, and machine learning-based classification, in the Methods section of the revised manuscript.

2. Is machine learning required for single molecule detection of glycans? If it is essential, the success of single molecule identification of glycans depends on the integration of hardware (tag introduction) and software (machine learning). If fusion is necessary, it should be introduced with appropriate papers in the introduction.

Response: We think machine learning method is a powerful tool to fuel nanopore sensing, especially when it comes to the identification of many analytes with minute structural difference. Besides, the introduction of machine learning method could minimize the risk of personal error in the analyte identification. In this work, the machine learning method we used is only a very preliminary proof-of-concept

experiment, which achieved the identification of glycans depending on the different scatter plot patterns derived from the processed blockage events. When the resolving power of nanopore towards oligosaccharide glycans is increasingly improved in the future, we can expect that the developed machine learning method can readily identify the glycan samples by directly distinguishing the waveform from the pristine current traces and by referring to the established reference library in the glycomics field. Therefore, we have added the brief description on why we use machine learning in nanopore sensing field with several typical reports (Ref. 24, 25, 41, 42). Considering the whole continuity of the Introduction section, the description was placed before the machine learning experiment in the revised manuscript as follows.

“... Inspection of the scatter plots also displays the difference among four glycans. However, the large overlap in scatter plots leads to the difficulty in unambiguously identifying glycans using human eye, particularly when it comes to the large number of analyte samples. To achieve the unequivocal identification of analytes according to nanopore data with subtle difference, machine learning-based methods have been increasingly explored as powerful supports⁴¹. Typical machine learning method is the employment of various classification algorithms that are used to discriminate and identify different analytes with minor difference in structure^{24, 42}, size⁴³, or charge²⁵ depending on the feature data extracted from either the waveforms²⁴ or the scatter plots²⁵. Here, we attempt to exploit the machine learning-based classification approach ...”

[24] Im, J., Lindsay, S., Wang, X. & Zhang, P. Single molecule identification and quantification of glycosaminoglycans using solid-state nanopores. *ACS Nano* **13**, 6308-6318 (2019).

[25] Xia, K., *et al.* Synthetic heparan sulfate standards and machine learning facilitate the development of solid-state nanopore analysis. *Proc. Natl. Acad. Sci. U.S.A.* **118**, e2022806118 (2021).

[41] Arima, A., Tsutsui, M., Washio, T., Baba, Y. & Kawai, T. Solid-state nanopore platform integrated with machine learning for digital diagnosis of virus infection. *Anal. Chem.* **93**, 215-227 (2021).

[42] Wang, Y., *et al.* Identification of nucleoside monophosphates and their epigenetic modifications using an engineered nanopore. *Nat. Nanotechnol.* **17**, 976-983 (2022).

[43] Taniguchi, M., *et al.* Combining machine learning and nanopore construction creates an artificial intelligence nanopore for coronavirus detection. *Nat. Commun.* **12**, 3726 (2021).

In addition, we have also made a discussion on the potential of machine learning method in the Discussions section of the revised manuscript as follows.

“... Most notably, the introduction of machine learning method in glycan identification experiments has significantly strengthened the identification ability of nanopore. This

proof-of-concept experiment based on only the scatter plot pattern foreshadows the great potential of the integration of nanopore sensing with machine learning method in future glycan analysis field. On one hand, as nanopore technique advances, both the number of glycan samples with similar structure, including small individual monosaccharide units, that need to be discerned, and the difficulty of artificial identification by nanopore data will inevitably increase. In this regard, machine learning method can be expected to assist nanopore sensing to achieve the unequivocal and rapid glycan identification only by small amount of nanopore data. On the other hand, our proof-of-concept experiment involves multiple isolated steps including nanopore recording, signal processing, and machine learning-based classification. Given the excellent data processing ability of machine learning⁴⁶, future integration of nanopore sensing with machine learning should be a fusion on a deeper level. The resultant artificial intelligent nanopore can be expected to directly and accurately recognize glycan analyte by identifying the waveform from the pristine current traces immediately after nanopore recording⁴⁷.”

[46] Wen, C., Dematties, D. & Zhang, S.-L. A guide to signal processing algorithms for nanopore sensors. *ACS Sens.* **6**, 3536-3555 (2021).

[47] Horejs, C. Artificially intelligent nanopore for rapid SARS-COV-2 testing. *Nat. Rev. Mater.* **6**, 650-650 (2021).

3. Lines 136 to 147 To confirm that the substance going through the nanopore is 6SL-MPB, the *trans* solution should be examined by mass spectrometry or NMR after 5 hours of monitoring. The current writing style leads one to believe that substances other than just 6SL-MPB are also mixed in the solution.

Response: We thank the Reviewer for the good suggestion. To better perform verification with MS, we have carried out a new translocation experiment involving much more AeL nanopores (80 ~ 90 nanopores) assembled in lipid bilayer membrane, a much higher analyte concentration (~ 100 μ M) to facilitate the translocation, and a much longer recording time (8.5 h). After that, the electrolyte solution in the *trans* compartment was collected. The collected solution was desalted with C18 SPE micro column (ACCHROM, UniElut C18, 200mg/3mL). Specifically, prior to use, the SPE column was washed with 3 mL 85% acetonitrile (v/v)/0.1%(v/v) trifluoroacetic acid (TFA) for activation. Then, 6 mL water/0.1%(v/v) TFA was used for the equilibration of the column. Then, after loading the sample into column, 6 mL water/0.1%(v/v) TFA was used to wash column to remove the salt. Finally, 3 mL 85% acetonitrile (v/v)/0.1%(v/v) TFA was

used to elute the sample. After freeze-drying, the purified sample was redissolved in only 20 μL water to perform MS test. The found m/z of 833.3181 verified the translocation of 6SL-MPB (calcd. for $[\text{M}+\text{H}]^+$ 833.3186) through AeL nanopore (Fig. R4d). We have included this result using a graphic (Supplementary Fig. 12) and a detailed description in the revised Supplementary Information and also made a brief description in the main text of the revised manuscript.

Fig. R4. Supplementary Fig. 12. The experiment **B** verifies the translocation of 6SL-MPB through AeL nanopore. **a**, Schematic of the translocation experiment through 80~90 AeL nanopores in lipid membrane. **b**, The current traces of 80~90 AeL nanopores with the translocation of 6SL-MPB at +100 mV. **c**, Graphic showing the C18-packed solid phase extraction (SPE) micro column. **d**, Graphic showing the used high resolution mass spectrometer: Agilent 6540 UHD quadrupole time-of-flight accurate-mass mass spectrometer. **e**, Partial mass spectrum of the concentrated sample showing the mass of 6SL-MPB (calcd. for $[\text{M}+\text{H}]^+$ 833.3186, found 833.3181).

4. Does the numerical representation in the confusion matrix in the text and the Supporting data represent the number of ionic current-time waveforms? You must explain what the numbers in the matrix mean.

Response: We apologize for the confusions here. The numerical representation in the confusion matrix represents the accumulative number of the test set in machine learning process. Specifically, in the case of four glycans (*i.e.*, 3SG-MPB, 3SL-

MPB, STetra2-MPB, and LSTa-MPB), 20 sets of feature data for each glycan were obtained. Accordingly, the feature matrix of four glycans was composed of 20×4 feature sets. All feature sets from each glycan represents a strata, which was randomly split to a 80% training set and a 20% test set. That is, 16 sets of features were randomly obtained as training sets to train the classifier model based on the selected algorithm. The remaining 4 sets of features, namely 4 test sets, were applied to test the trained classifiers and ensure reliability. After one cycle of training and test process, 4 test sets from each glycan were predicted. This cycle was repeated 100 times. Thus, the prediction of 400 test sets from each glycan was completed, which constituted an experiment. Finally, the prediction result of 4000 test sets was obtained after 10 replicates to plot the confusion matrixes.

We have carefully detailed the description of the machine learning part in the Method section. And we have added the detailed description on the numerical representation in the confusion matrix in the corresponding figure legend in the revised manuscript as follows.

“... **g**, Evaluation results of five models in terms of F1, Precision, and Recall scores. These scores were the mean values of 10 replicates, each replicate consists of 100 training and test cycles. **h**, Confusion matrix of SVM model from the accumulated prediction result of 4000 test sets after 10 replicates. One training and test cycle includes the prediction result of 4 test sets of feature data of each glycans. ... **k**, Confusion matrix of SVM model from the accumulated prediction result of 4000 test sets after 10 replicates. ...”

5. The numbers in the confusion matrix are different for each of the three sorts of molecules. For instance, in Figure 2h, they are 3,000, 3,000, and 4,000. Usually, in machine learning, the number of data for each glycan is kept constant to prevent bias. Here everyone should converge on 3,000.

Response: We apologize for this oversight. We have re-perform all machine learning classification calculations based on the identical event number of each glycan sample to avoid bias.

Reviewer #3 (Remarks to the Author):

This article describes the use of nanopores to detect and discriminate modified small glycans. The text is globally clear and mostly convincing. The experiments and interpretations are superficial. Furthermore, this research lacks somehow of novelty: the glycans used are well discriminated with chemical modification and standard analytical methods. Thus, we do not see the added value of the nanopore detection. Furthermore, the use of chemical modifications removes the interesting ability of nanopores to distinguish molecules without tagging.

Response: We are extremely grateful for the Reviewer's comments and criticisms. The Reviewer's expertise in nanopore field has helped us greatly to improve the manuscript and also provided guidance to our future study work on nanopore sensing. According to the Reviewer's professional suggestions, we have strengthened this work through additional experiments and discussions.

On the novelty of nanopore glycan sensing, we want to further clarify our views.

Glycans are one of the four fundamental classes of macromolecules that comprise living systems, along with nucleic acids, proteins, and lipids, and play the central role in almost every biological process. The extreme complexity in the structure of glycans derived from a large number of naturally occurring monomers (about hundreds of monosaccharides when including plant and bacterial glycans) and **the diversity in linkage, isomerism, and branching**, and the extremely low abundance, seriously challenge conventional analysis techniques. Sometimes, **the issue of glycan linkage isomer distinction cannot even be resolved without the combination of multiple tools**, for example the combination of mass spectrometry and liquid chromatography or capillary electrophoresis or ion mobility spectrometry. Thus, the National Research Council (US) Committee on Assessing the Importance and Impact of Glycomics and Glycosciences in a report of *Transforming Glycoscience: A Roadmap for the Future* made a finding that "a suite of widely applicable tools, analogous to those available for studying nucleic acids and proteins, is needed to detect, describe, and fully purify glycans from natural sources and then to characterize their chemical composition and structure"¹. In this regard, nanopore-based sensing technique that can deal with sample diversity and low abundance with single-molecule sensitivity provides an extremely competitive tool. Also, as Prof. Jason R. Dwyer said², glycans are emerging as the latest high-

profile targets of nanopore sensing technique. Consequently, nanopore sensing technique can be expected to make a contribution to glycomics and glycoscience that could be beyond its contribution to genomics or proteomics where the existing analysis techniques leave less of a gap between aims and achievability.

A number of exquisite and informative reports have achieved the polysaccharide nanopore measurement, forming important touchstones for the consideration of nanopore sensing as a general tool for glycan analysis. However, these few reports are largely confined to those large glycan (polysaccharide) molecules. On the contrary, numerous small oligosaccharide glycans consisting of 1 to 20 monosaccharide building blocks that attached to protein surface or in free form are at the very core of glycomics analysis. Judging from the current situation, it seems that nanopore sensing is at the end of its ability in the face of these small glycans, which might be enslaved to the much smaller sizes of oligosaccharide glycans and/or their much lower charge density.

Given that the derivation of glycans is a widely used strategy in various glycan analyses methods (see the next response for more details), we in this manuscript adopted the similar derivation strategy, which addressed the issue of the small size of oligosaccharide molecules and the low interaction affinity with nanopore interface and thus achieved the nanopore detection of a number of oligosaccharide glycans. Put in stark terms, the glycan derivation with a tag indeed weakens the power of nanopore sensing to some extent, one major advantage of which is label-free detection. Therefore, the following important task is to explore and achieve the nanopore-based label-free glycan detection by protein engineering and chemical modification of nanopore protein. Only then will we be close to the goal of glycan profiling with nanopore and can further advance the more challenging glycan nanopore sequencing.

In the revised manuscript, we have also added some discussion on the significance of nanopore glycan sensing and the focus of the following work (achieving the label-free detection towards the native glycans) in the Discussion section. We believe that this revised manuscript will be of great interest to researchers of glycoscience field working on glycan structure analysis and characterization.

Refs:

- [1] The National Academies of Sciences Consensus Report. Transforming glycoscience: a roadmap for the future. Washington, DC: The National Academies Press; 2012.

[2] Hagan, J. T., *et al.* Chemically tailoring nanopores for single-molecule sensing and glycomics. *Anal. Bioanal. Chem.* **412**, 6639 (2020).

The chemical modifications. Although the chemistry used is clearly mastered, the presented results raise some questions and remarks:

1. What did bring the author to use benzene derivates which are known to be very toxic? Why didn't they use other common molecules such as fluorochromes?

Response: One of big challenges of glycan analysis faced is the lack of chromophores or fluorophores. Thus, many routine analytical approaches depend on the chemical modifications of glycans. For example, permethylation allows facile gas phase analyses by improving thermal stability and volatility. Introduction of chromophores or fluorophores can increase the sensitivity and detectability of some analytical techniques like chromatography, capillary electrophoresis, and mass spectrometry. Thus, the exposed single reducing terminus of glycans has been widely exploited to link chromophores or fluorophores or charged moieties to render the glycans amenable to various analytical techniques. The most common modification reaction employed is the reductive amination, where a label containing a primary amine group reacts in a condensation reaction with the aldehyde group of the glycan, resulting in a Schiff base, which is reduced by a reducing agent to yield a secondary amine. This modifying approach has the advantages of high efficiency and the stoichiometric attachment of one label per glycan.

The most widely used chromophores or fluorophores include 2-aminobenzamide (2-AB), 2-aminobenzoic acid (2-AA), 2-aminopyridine (PA), 7-amino-4-methylcoumarin (AMC), 2-aminonaphthalene trisulfonic acid (ANTS), and 1-aminopyrene-3,6,8-trisulfonic acid (APTS). These molecules share the same aromatic core. On the other hand, the literatures also showed us that the cationic residues from Lys (K238 and K242) and Arg (R220 and R282) of wild-type AeL constitute the key recognition sites of nanopore. All these factors promoted us to introduce aromatic elements into glycan molecules, which might be able to increase the interaction of the corresponding glycan derivative with nanopore recognition sites. Furthermore, many commercially available oligosaccharide samples (*e.g.*, Gb3- β -MP) often present as the form of glycosides with a 4-methoxyphenyl group. Therefore, at the beginning of this work, we attempted to introduce the 4-methoxyphenyl group through the reductive amination between the reducing ends

of glycans and *p*-anisidine. Then, the scope of tag molecules was expanded around the phenylamine core.

As for other fluorochrome molecules, there might be a better molecule as the tag of glycans to produce more remarkable and distinguishable blockage signals in nanopore test, which needs to explore in the following work.

2. The chemistry part and the analysis of the reactions are clear. I have nevertheless one concern: The author should explain how they measured the concentration of tagged molecules.

Response: The concentration (or purity) of tagged glycan molecules was determined by performing the peak area analysis of HPLC spectrum, as shown in Fig. R5. HPLC purity assay by calculating percentage of peak area in relation to total area of peaks is often used to estimate the concentration (purity) of analyte. We have added a detailed description on the purity assay in the Method section as follows.

“... The purity of glycan derivatives was determined by carrying out the peak area analysis of HPLC spectrum, which calculates the percentage of peak area in relation to total area of peaks. The general chromatographic conditions: BOSTON Green ODS column (4.6 mm × 250 mm, 10 μm), room temperature, binary gradient elution (eluent A: H₂O with 0.1% TFA(v/v), eluent B: CH₃CN with 0.1 %TFA(v/v)), flow rate of 1 mL·min⁻¹.”

==== Shimadzu LabSolutions 分析报告 ====

Fig. R5. HPLC analysis report of 6SL-MPB. The chromatographic condition: BOSTON Green ODS column (4.6 mm × 250 mm, 10 μm), flow rate of 1 mL·min⁻¹, eluent A: H₂O with 0.1% TFA(v/v), eluent B: CH₃CN with 0.1 %TFA(v/v), 10%–90% elute B, 12 min.

3. The type of chemical modifications used in this paper are too heavy to be used in the separation of glycans.

Response: We agree the Reviewer that the tag molecule we currently used is a little big. However, the incorporation of such big tag is to produce the prominent and distinguishable nanopore blockage signals. Our test results have shown that the incorporation of relatively small tag molecule into glycans only elicited weak current blockages based on the currently used wild-type AeL nanopore, in which case the difference in nanopore data among glycans is hard to discern. Our following work is focusing on the improvement of resolving power of AeL nanopore by site-specific mutagenesis and chemical modification. We aim to achieve the label-free detection of oligosaccharide glycans based on the engineered protein or chemically modified protein or solid-state nanopore.

4. The chemistry for neutral glycans is even heavier and gives rise to pentasaccharides and not disaccharides as for the other experiments.

Response: We agree the Reviewer that the composite tag what we used for neutral glycans is much bigger than that for sialylglycans. And we also recognized that such big tag molecule might conceal the target glycan to some extent in the nanopore signal, leading to the slight differences among neutral glycans (*e.g.*, three disaccharide isomers). In light of this, we have been trying to improve the type of tag molecule with an aim of achieving the electrophoresis-driven translocation of neutral glycans through nanopore. In this respect, the Reviewer#1 also advise us to try out 3-(4-aminophenyl)propionic acid or 4-(4-aminophenyl)butyric acid as the tag of the neutral glycans, although both of them also did not work. Yet there are some encouraging signs that the trisaccharide unit in the composite tag can be replaced with an acidic monosaccharide, for example, glucuronic acid or glucose-6-phosphate, which thus can offer a relatively small composite tag to neutral glycans. And we have now obtained some preliminary results. Anyway, the great efforts should be made to explore and achieve the label-free detection of glycans regardless of acidic or neural analytes.

The nanopore experiments. The conditions used in this paper are very conventional except for certain subtleties.

1. The low pass filter cutoff frequency set for the single-channel recordings is extremely low. Why use a 5kHz when most of the timescales measured are smaller than 1 ms. The consequence of a so low cutoff is a distortion of the event cloud representation which can be clearly seen on each graph of the paper.

Response: The low-pass filtering frequency of 5kHz was used in our nanopore measurements by referring to the reported literature (Cao C., *et al.* Discrimination of oligonucleotides of different lengths with a wild-type aerolysin nanopore. *Nat. Nanotech.* 2016, 11, 713) where oligonucleotides (dA₂ in particular) are close to oligosaccharides in size. As the Reviewer has indicated, the low cut-off frequency, although it can reduce the noise of the signal, can indeed lead to the distortion of the translocate events, especially those short event with higher blockade current. Thus, we have evaluated the effect of low-pass filtering frequency on the nanopore data by taking the translocation experiment of 6SL-MPB as an example. Figure R6 shows the test results acquired at the 250kHz sampling rate and varying low-pass filtering frequencies. We can see that the very low frequency, like 1kHz or 500 Hz, weakens the blockage signals according to the current traces, results in the increase

of the number of events with duration of ~ 0.2 ms and I_b/I_0 of ~ 0.6 and the severe distortion of event scatter plot. Under a higher low-pass frequency, like 50kHz, the characteristic distribution (~ 0.42 in I_b/I_0) appears more prominent. It should be noted that nanopore data acquired at filtering frequency of 5kHz is mostly identical to those acquired at filtering frequency of 50kHz, except for some short events with around 0.2 ms in dwell time. Therefore, in the following nanopore studies, we will adopt the proper low-pass frequency by comprehensively considering the analyte type and the blockage signal characteristic.

Fig. R6. Representative AeL nanopore current trances, scatter plot of I_b/I_0 vs. dwell time, and the corresponding I_b/I_0 and dwell time distribution of 6SL-MPB acquired using a 250kHz sampling rate and varying low-pass filtering frequencies. Each scatter plot contains at least

9,000 events.

The words "single molecule discrimination" are misleading. Even though the clouds of events have a different shape for each probed molecule, they all are located in the same region of the Dwell times vs blocked current plots. An event with a dwell time of 0.2 ms and with a blocked current of 0.6 can be seen in each cloud representation of the paper !!!

Response: We agree with the Reviewer and apologize for the less rigorous statement "single molecule discrimination/identification". We have modified the Title of our manuscript to "Glycan identification with a protein nanopore" and the related description to avoid any misleading implications. Nevertheless, we still look forward to in the near future achieving the identification of glycan at the single-molecule level by recognizing each waveform by using some strategies, protein engineering, for example, to increase the resolving power of protein nanopore.

Mixtures of two or more molecules are necessary to prove the discrimination.

Response: We appreciate this suggestion. In the preceding response, we, according to the Reviewer#1's suggestion, have tested the larger glycans, four pentasaccharide isomers (e.g., LSTa, LSTd, LSTc, and LSTb) with MPB tag. The nanopore test results show the distinct difference among four glycans. Furthermore, we designed a measurement towards the mixtures of these glycans based on a same AeL nanopore by adding the glycan sample into *cis* solution in sequence (Fig. R7). The results show that LSTb and LSTd were readily recognized in different mixtures due to their prominent and distinct characteristic. As for LSTa and LSTc, these two glycans can be roughly recognized when the mixture contain fewer samples, for example, LSTa-LSTb mixture and LSTa-LSTb-LSTc mixture. When the mixture contains all four glycans, LSTa and LSTc are basically indistinguishable due to severe superposition of scatter plots.

Fig. R7. (a-d) Scatter plots of four pentasaccharide isomers, LSTa-MPB (a), LSTb-MPB (b), LSTc-MPB (c), and LSTd-MPB (d), and the corresponding chemical structure. (e-h) Scatter plots and the corresponding I_b/I_0 distributions acquired when LSTa-MPB, LSTb-MPB, LSTc-MPB, and LSTd-MPB were sequentially added to the *cis* solution. All measurements were done in a 10 mM Tris-HCl buffer containing 1 mM EDTA and 1 M KCl with pH 8.0 at +100 mV potential applied to the *trans* side. All nanopore data were recorded using a 250 kHz sampling rate with a 5 kHz low-pass filtering.

We admit that some glycan samples cannot be differentiated when mixed together based on the current nanopore test strategy. The ultimate reason is that some glycan molecules are too small in structure to produce the outstanding blockage signals in AeL nanopore. To address this, our following work has focused on the AeL engineering modification of AeL and MspA, including the site-directed mutagenesis and chemical post-modification of inner wall, to further decrease the pore size to finally improve the resolving power of these nanopores to small glycan molecule. We are committed to achieve the label-free detection and recognition of these oligosaccharide glycans with engineered nanopores, as well as the glycan profiling towards all protein-attached glycans.

The concentration of glycans is extremely low compared to the one usually found in the literature. The event frequency is not mentioned in the text of the paper but must be very low as well (a couple of Hz according to Figure 1f). Therefore, the experiment where the *trans* side of the setup is analyzed by putting it back in the *trans* is absolutely

not convincing. With such an event frequency, the translocated molecules would have to be accumulated during years in the trans side to reach enough concentration even to get an event frequency of 1 mHz. The author should explain the details of this experiment.

Response: Prior to nanopore recording, glycan (taking 6SL derivative as an example) solution (2 mM) of 1 μL was added in the electrolyte solution ($\sim 1\text{mL}$) of the *cis* compartment. The final glycan concentration is approximately 2 μM . The produced blockage signal frequency we calculated is approximately 13.7 Hz (the corresponding representative current traces shown in Fig. 1f). To perform the translocation verification experiment, we first attempted to assemble multiple AeL nanopores in the lipid bilayer membrane by injecting large amount of AeL protein monomer solution (about fourfold the amount for the single nanopore insertion). Finally, we obtained 7 or 8 AeL nanopores (from the estimation according to the open current) inserted in the lipid bilayer membrane. Then, 20 μL 2 mM 6SL-MPB solution was added into the electrolyte solution of the *cis* compartment, the final concentration was approximately 40 μM . Under the applied voltage of +100 mV, the glycan molecule translocated continually through multiple AeL nanopores. The recorded representative current traces are shown in Fig. R8b. The translocation experiment lasted about 5 hours at +100 mV (During this period, small amount of electrolyte solution was added to the *cis* and *trans* compartments to compensate the volatilized liquid). Then, the experiment was stopped, the solution in the *trans* compartment was collected. Then, a new experimental set-up was assembled based on a perfusion chamber and a perfusion cup, where the corresponding *trans* compartment was added with the new electrolyte solution, while the *cis* compartment was added with the collected solution from the above experiment. Upon the formation of lipid bilayer and subsequent successful insertion of a single AeL nanopore, the ionic current was recorded at +100 mV immediately. The whole recording lasted ~ 67 minutes, which is shown in Fig. R8c. After signal analysis, 442 effective blockage events from 67 minutes' recording were extracted, as shown in the scatter plots of Fig. R8d. We can observe the characteristic population in the scatter plot that corresponds to the 6SL-MPB through the comparative analysis with the nanopore data of 6SL-MPB. This suggests the occurrence of glycan translocation through AeL nanopore.

Thus, we think that by taking advantage of multiple nanopores, higher concentration analyte solution and long translocation time under applied voltage,

detecting the analyte molecule in the *trans* solution by adding it in the *cis* compartment of a newly-assembled nanopore set-up is feasible. To further detail our experiment, we have included the representative current traces of 7 or 8 AeL nanopores under the applied voltage of +100 mV (Fig. R8b). And we have also presented all current traces with time scale from 67 minutes' recording (Fig. R8c), instead of exhibiting several representative current traces without time scale. Besides, we also added more details on the experiment process. Finally, we apologize for the misunderstanding caused by our negligence in results' representation and the relevant description.

Fig. R8. Supplementary Fig. 11. The designed experiment A verifies the translocation of glycan

derivative through AeL nanopore. **a**, Schematic of the experimental process. **b**, The representative nanopore ionic current traces of ~7 AeL nanopores with the translocation of 6SL-MPB at +100 mV. **c**, All nanopore ionic current traces based on a newly assembled AeL nanopore by adding the collected *trans* solution. The whole recording lasted nearly 67 minutes. **d**, The scatter plot of the recorded blockage signals.

Moreover, we have conducted additional translocation experiment to detect the translocated glycan analyte using mass spectrometry (Fig.R9). This experiment involves much more AeL nanopores (80 ~ 90 nanopores) assembled in lipid bilayer membrane, a much higher analyte concentration (~ 100 μ M) to facilitate the translocation, and a much longer recording time (8.5 h). After that, the electrolyte solution in the *trans* compartment was collected. The collected solution was desalted with C18 SPE micro column (ACCHROM, UniElut C18, 200mg/3mL). Specifically, prior to use, the SPE column was washed with 3 mL 85% acetonitrile (v/v)/0.1%(v/v) trifluoroacetic acid (TFA) for activation. Then, 6 mL water/0.1%(v/v) TFA was used for the equilibration of the column. Then, after loading the sample into column, 6 mL water/0.1%(v/v) TFA was used to wash column to remove the salt. Finally, 3 mL 85% acetonitrile (v/v)/0.1%(v/v) TFA was used to elute the sample. After freeze-drying, the purified sample was redissolved in only 20 μ L water to perform MS test. The found *m/z* of 833.3181 verified the translocation of 6SL-MPB (calcd. for $[M+H]^+$ 833.3186) through AeL nanopore (Fig. R9d). We have included this result using a graphic (Supplementary Fig. 12) and a detailed description in the revised Supplementary Information.

Fig. R9. Supplementary Fig. 12. The experiment **B** verifies the translocation of 6SL-MPB through AeL nanopore. **a**, Schematic of the translocation experiment through 80~90 AeL nanopores in lipid membrane. **b**, The current traces of 80~90 AeL nanopores with the translocation of 6SL-MPB at +100 mV. **c**, Graphic showing the C18-packed solid phase extraction (SPE) micro column. **d**, Graphic showing the used high resolution mass spectrometer: Agilent 6540 UHD quadrupole time-of-flight accurate-mass mass spectrometer. **e**, Partial mass spectrum of the concentrated sample showing the mass of 6SL-MPB (calcd. for $[M+H]^+$ 833.3186, found 833.3181).

It has been shown in the literature that a higher salt concentration and lower voltage are needed for nanopore detection. The author should provide data comparing their conditions and the ones found in the literature.

Response: When we first performed the nanopore test toward negatively charged 6SL or 3SL, or LSTa, we found there was no evidential blockage signals in the recorded current traces. And we also found in the reported literatures that higher applied voltage, higher analyte concentration, or higher salt concentration can significantly increase the current blockage frequency. For example, a report from Prof. Juan Pelta's group¹ showed that the events frequency of dextran sulfate transported through AeL nanopore increases exponentially as a function of applied voltage and linearly as a function of dextran sulfate concentration. A work from Prof. Xiyun Guan's group² reported the increase of the event frequency of a peptide

traversing hemolysin nanopore was observed when the concentration of NaCl electrolyte solution from 1M to 3M. Thus, we attempted to increase the applied voltage (from +60 mV to +160 mV), the salt concentration (from 1M KCl to 4M KCl), and the analyte concentration (from 2 μ M to 10 μ M) to explore whether the obvious blockage signals could be observed. After these, we still cannot observe the obvious blockage signals from 3SL, 6SL, or LSTa. Thus, the statement of Supplementary Fig. 2' comment and the citing reference in the original manuscript might resulted in misunderstanding. We apologize for the misunderstanding. We have modified the statement by combining the Supplementary Fig. 2 and 3, and have cited the above literatures.

Refs.

- [1] Pastoriza-Gallego, M., *et al.* Dynamics of a polyelectrolyte through aerolysin channel as a function of applied voltage and concentration. *Eur. Phys. J. E* **41**, 58 (2018).
- [2] Chen X., *et al.* Salt-mediated nanopore detection of ADAM-17. *ACS Appl. Bio Mater.* **2**, 504–509 (2019).

The author should provide the experimental condition in all figure legends.

Response: We thank Reviewer's suggestion and have now added the experimental condition in all figure legends.

The representation of the current traces is the one that was proposed in the literature for solid-state nanopores for which only the drop of conductance is important because the baseline current is pore dependent. Here with a protein nanopore, the baseline current is an important control. For instance, in SFig 5 it seems that the 2 pores are not used at the same voltage... The author should provide the zero current level on each trace.

Response: We thank the Reviewer's suggestion and have replotted all the current traces with the zero current level. As for Supplementary Fig. 5, both current traces were recorded with the same applied voltage of +100 mV, the difference between them is derived from the difference in noise level.

The comment on page 24 of the Supp Information file underlines the fact that the authors seem to not master completely the nanopores experiments. The molecules do not enter the pore or cannot be seen?

Response: We apologize for the ambiguity of the original manuscript. What we want to say is that the translocation of glycan (3SL or 6SL) derivative with straight-chain alkane as tag through AeL nanopore can not be sensed with AeL nanopore, which thus cannot be reflected in the blockage signals. We have now modified the statement as follows.

“..., implying that the translocation of 3SL or 6SL derivative with straight-chain alkane as tag through AeL nanopore cannot be sensed.”

Machine learning. As I am not a specialist in these analysis techniques, I can just say that the procedure of ML is not enough detailed to fully understand what is done and what it is done for.

Response: We apologize for the unclear description of the original manuscript. We have added full experimental details on the use of ML method in Method section and detailed elucidation on the graphics of the ML results. In addition, we think machine learning method is a powerful tool to fuel nanopore sensing, especially when it comes to the identification of some analytes with minute structural difference, where the differences among analyte's signals are small. The introduction of machine learning method can minimize the risk of personal error in the analyte identification. Therefore, we have added the brief description on the use of machine learning in nanopore sensing field with several typical references before the machine learning experiment in the revised manuscript as follows.

“... Inspection of the scatter plots also displays the difference among four glycans. However, the large overlap in scatter plots leads to the difficulty in unambiguously identifying glycans using human eye, particularly when it comes to the large number of analyte samples. To achieve the unequivocal identification of analytes according to nanopore data with subtle difference, machine learning-based methods have been increasingly explored as powerful supports⁴¹. Typical machine learning method is the employment of various classification algorithms that are used to discriminate and identify different analytes with minor difference in structure^{24, 42}, size⁴³, or charge²⁵ depending on the feature data extracted from either the waveforms²⁴ or the scatter plots²⁵. Here, we attempt to exploit the machine learning-based classification approach ...”

Finally, we have also discussed the potential and development of ML in the

nanopore sensing towards glycans or glycomics analysis in Discussions section as follows.

“... Most notably, the introduction of machine learning method in glycan identification experiments has significantly strengthened the identification ability of nanopore. This proof-of-concept experiment based on only the scatter plot pattern foreshadows the great potential of the integration of nanopore sensing with machine learning method in future glycan analysis field. On one hand, as nanopore technique advances, both the number of glycan samples with similar structure, including small individual monosaccharide units, that need to be discerned, and the difficulty of artificial identification by nanopore data will inevitably increase. In this regard, machine learning method can be expected to assist nanopore sensing to achieve the unequivocal and rapid glycan identification only by small amount of nanopore data. On the other hand, our proof-of-concept experiment involves multiple isolated steps including nanopore recording, signal processing, and machine learning-based classification. Given the excellent data processing ability of machine learning⁴⁶, future integration of nanopore sensing with machine learning should be a fusion on a deeper level. The resultant artificial intelligent nanopore can be expected to directly and accurately recognize glycan analyte by identifying the waveform from the pristine current traces immediately after nanopore recording⁴⁷.”

REVIEWERS' COMMENTS

Reviewer #1 (Remarks to the Author):

As suggested by reviewer 1 the authors have examined:

1. LSTa vs LSTb vs LSTc vs LSTd
2. Lea vs Lex
3. LNT vs LNnT

The results indicate that a clear difference can be observed between the first two sets of isomers, whereas with LNT vs LNnT the difference is marginal (as we anticipated), such that the I_b/I_0 values are very close but the densities and spreading of the dots are different.

Thus It is convincing that indeed isomers can be distinguished using their nanopore method. These data have been included in the main text and figures.

Additionally the authors have tested the use of a charged benzyl linker instead of a sialic acid containing tag. However this linker cannot induce any signal when passing the nanopore. The authors suggest that although the carboxylic acid is located some distance from the benzene ring, it is too small to induce the blockage of nanopore.

In summary, the authors have addressed constructively our suggestions, with pretty convincing results.

Reviewer #2 (Remarks to the Author):

The authors fully improved the manuscript and addressed my questions satisfactorily. I think that the manuscript is ready for publication in Nature Communications.

Reviewer #1 (Remarks to the Author):

As suggested by reviewer 1 the authors have examined:

1. LSTa vs LSTb vs LSTc vs LSTd
2. Lea vs Lex
3. LNT vs LNnT

The results indicate that a clear difference can be observed between the first two sets of isomers, whereas with LNT vs LNnT the difference is marginal (as we anticipated), such that the I_b/I_0 values are very close but the densities and spreading of the dots are different.

Thus, it is convincing that indeed isomers can be distinguished using their nanopore method. These data have been included in the main text and figures.

Additionally the authors have tested the use of a charged benzyl linker instead of a sialic acid containing tag. However this linker cannot induce any signal when passing the nanopore. The authors suggest that although the carboxylic acid is located some distance from the benzene ring, it is too small to induce the blockage of nanopore.

In summary, the authors have addressed constructively our suggestions, with pretty convincing results.

Response: We thank the Reviewer for the positive comment to our manuscript, which will urge us to continue move forward in nanopore glycan sensing filed.

Reviewer #2 (Remarks to the Author):

The authors fully improved the manuscript and addressed my questions satisfactorily. I think that the manuscript is ready for publication in Nature Communications.

Response: We appreciate that the Reviewer supports the publication of this manuscript in Nature Communications